# CFD: Learning Generalized Molecular Representation via Concept-Enhanced Feedback Disentanglement

**Aming Wu, Cheng Deng**[*]
School of Electronic Engineering, Xidian University, Xi'an, China
amwu@xidian.edu.cn, chdeng@mail.xidian.edu.cn

## Abstract

To accelerate biochemical research, e.g., drug and protein discovery, molecular representation learning (MRL) has attracted much attention. However, most existing methods follow the closed-set assumption that training and testing data share identical distribution, which limits their generalization abilities in out-of-distribution (OOD) cases. In this paper, we explore designing a new disentangled mechanism for learning generalized molecular representation that exhibits robustness against distribution shifts. And an approach of Concept-Enhanced Feedback Disentanglement (CFD) is proposed, whose goal is to exploit the feedback mechanism to learn distribution-agnostic representation. Specifically, we first propose two dedicated variational encoders to separately decompose distribution-agnostic and spurious features. Then, a set of molecule-aware concepts are tapped to focus on invariant substructure characteristics. By fusing these concepts into the disentangled distribution-agnostic features, the generalization ability of the learned molecular representation could be further enhanced. Next, we execute iteratively the disentangled operations based on a feedback received from the previous output. Finally, based on the outputs of multiple feedback iterations, we construct a self-supervised objective to promote the variational encoders to possess the disentangled capability. In the experiments, our method is verified on multiple real-world molecular datasets. The significant performance gains over state-of-the-art baselines demonstrate that our method can effectively disentangle generalized molecular representation in the presence of various distribution shifts. The source code will be released at https://github.com/AmingWu/MoleculeCFD.

## 1 Introduction

With the rejuvenation of deep neural networks, to advance the development of biochemical research, e.g., drug discovery (Sanchez-Fernandez et al., 2022) and protein design (Ingraham et al., 2019), molecular representation learning (MRL) has attracted growing attention, which aims to transform molecules into low-dimensional and dense vectors (Gilmer et al., 2017; Yang et al., 2019; Rong et al., 2020; Fang et al., 2022; Yang et al., 2022; Zhuang et al., 2023). By means of the learned molecular representation, many downstream tasks could be addressed effectively, including drug property prediction (Wang et al., 2024b), binding affinity analysis (Karimi et al., 2019), and search of antibiotics (Stokes et al., 2020), etc.

Though MRL has achieved significant progress, most existing methods often follow the closed-set assumption, i.e., the training and testing data share the same distribution. However, in practical biochemical research, molecules may exhibit many variants and originate from different distributions (Ji et al., 2022; Mendez et al., 2019). Taking the virtual screening (Lim et al., 2019) as an example, the model may be affected by the distribution shift in the molecule itself, e.g., size (Ji et al., 2022) or scaffold (Wu et al., 2018) changes. Furthermore, some unpredictable events like COVID-19 may bring new target molecules from unknown distributions. Obviously, these scenarios pose significant

---

[*]Corresponding Author

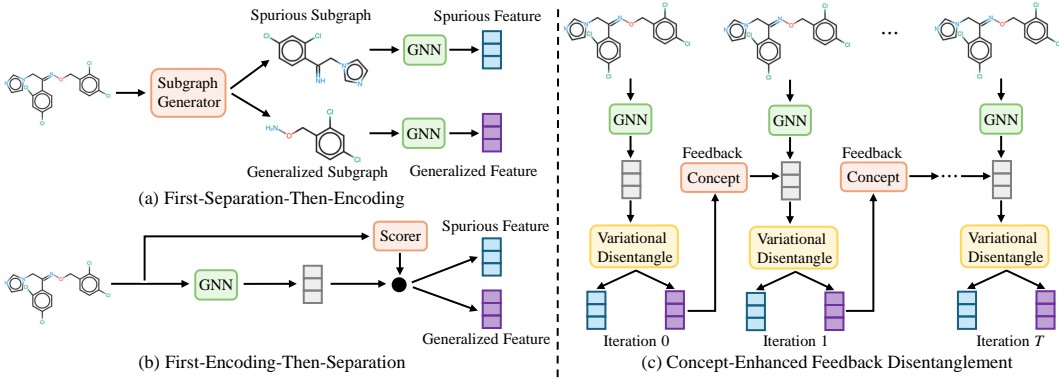

Figure 1: Three different mechanisms for disentangling generalized features. (a) and (b) are commonly used disentangled methods (Zhuang et al., 2023). (a) first uses a subgraph generator to separate the input, then each subgraph is encoded, respectively. (b) first encodes the input, then the representation is separated by a scorer (Zhuang et al., 2023). (c) is our proposed mechanism for learning generalized representation, i.e., Concept-Enhanced Feedback Disentanglement, which exploits the iteration manner to perform variational disentangled operations based on the feedback received from the previous iteration's output. The combination of all decomposed generalized features is taken as the final generalized representation for downstream tasks.

challenges for closed-set assumption based MRL methods. To this end, improving the generalization of molecular representation is meaningful for reducing the impact of distribution shifts.

Existing methods (Ahuja et al., 2021; Yang et al., 2024) mainly focus on learning generalized representation for regular Euclidean data (e.g., images). Instead, to preserve rich structural information (Gilmer et al., 2017), molecules are often represented as graphs, a typical non-Euclidean data, where atoms and bonds are separately taken as nodes and edges. Thus, the methods (Ahuja et al., 2021; Yang et al., 2024) corresponding to Euclidean data could not be directly applied to molecular graph data. Particularly, the work (Ji et al., 2022) points out that existing OOD methods (Sagawa et al., 2019; Sun & Saenko, 2016; Zhang, 2017) fail to significantly improve the performance of MRL tasks against distribution shifts. In this paper, we explore designing a disentangled mechanism for learning generalized molecular representation.

Recently, some studies (Chen et al., 2022; Li et al., 2022b; Yang et al., 2022) have attempted to improve the generalization of molecular representation, which could be classified into two types, i.e., First-Separation-Then-Encoding and First-Encoding-Then-Separation (Zhuang et al., 2023). Specifically, as shown in Fig. 1 (a), the first type first divides the graph into generalized and spurious subgraphs and then encodes each part separately. However, this kind of method usually relies on the separation quality. For certain molecules involving complex structures, since there is no high-quality decomposition, the performance of these methods may be weakened severely. To this end, the work (Zhuang et al., 2023) introduces a "First-Encoding-Then-Separation" mechanism. As shown in Fig. 1 (b), it first employs a Graph Neural Network (GNN) (Kipf & Welling, 2017; Hamilton et al., 2017) to encode the molecule. Then, another GNN is utilized to calculate the score of molecule representation, which is taken as the critical factor for separating generalized and spurious features. However, for extremely complex and entangled molecules, e.g., biomacromolecules, only leveraging the calculated score could not effectively obtain expected separation results. Meanwhile, this method (Zhuang et al., 2023) fails to focus on certain important substructures containing invariant characteristics, affecting its performance in real-world molecules.

In Fig. 1 (c), we propose a new disentangled mechanism, i.e., Concept-Enhanced Feedback Disentanglement (CFD), which still follows the "First-Encoding-Then-Separation" strategy and exploits the feedback mechanism (Zamir et al., 2017) to learn generalized representation. Concretely, given an input molecule, we first exploit a GNN to encode its representation. Then, two dedicated variational encoders are designed to separately decompose distribution-agnostic generalized features and spurious features. Next, a series of concepts are learned to capture certain critical substructures involving invariant molecule characteristics. By fusing these concepts into the decomposed distribution-agnostic features, the generalization ability could be further strengthened. Later, we iteratively perform the above disentangled operations based on a feedback received from the previous iteration's output.

Finally, based on the outputs of multiple feedback iterations, a self-supervised objective is constructed to encourage the variational encoders to own the disentangled capability. Extensive experimental results on multiple molecular datasets demonstrate the superiorities of our method.

The contributions are summarized as follows:

(1) We propose a paradigm of feedback disentanglement that utilizes the iteration manner to decompose generalized molecular representation, improving the robustness against distribution shifts.

(2) We propose a concept mining module focusing on certain critical substructures involving invariant characteristics. By fusing these concepts into the disentangled representation, the generalization ability could be further enhanced.

(3) We conduct extensive experiments on a diverse set of real-world molecule datasets (Zhuang et al., 2023) with various distribution shifts. The significant performance gains over multiple baselines demonstrate that our method can disentangle generalized molecular representation effectively. Moreover, to further verify the ability of capturing important substructure, our method is evaluated on a new task, i.e., Molecule's Ground-State Conformation Prediction (Xu et al., 2024). We observe that plugging our method into Transformer architecture (Xu et al., 2024) could still improve the prediction performance, indicating the superiorities of our method.

## 2 RELATED WORK

**OOD Generalization.** Recent research (Wang et al., 2022a; Robey et al., 2021; Zhou et al., 2024) has indicated that existing machine learning methods usually suffer from significant performance degradation under distribution shifts, motivating a surge of works on OOD generalization. Particularly, most existing methods (Evuru et al., 2024; Ratner et al., 2017; Feder et al., 2024) attempt to improve the generalization from the data and representation perspectives. Particularly, an effective strategy is to utilize data augmentation to improve the generalization. Zhou et al. (Zhou et al., 2020) utilize domain information for creating an additive noise to increase the diversity of training data distribution while preserving the semantic information of data. Xu et al. (Xu et al., 2021a) exploit the classical Fourier transformation to obtain the augmented images with diverse styles, which enhances the generalization ability effectively. Besides, by simply mixing up different styles of input data, the work (Zhou et al., 2021) could strengthen the generalization performance significantly.

Another effective strategy is to learn invariant representations (Chang et al., 2020; Creager et al., 2021). In general, the learned invariant representation should satisfy such a principle: 1) sufficiency: shows sufficient predictive power for the target, 2) invariance: contributes to optimal performance for the downstream tasks across all environments. Therefore, some works (Mahajan et al., 2021; Lv et al., 2022; Wang et al., 2022b) explore learning invariant representation from the causal perspective. Furthermore, feature disentanglement (Li et al., 2022a; Zhang et al., 2022; Wu et al., 2021) is also a commonly used mechanism to decompose distribution-agnostic representations. Though these methods have been demonstrated to be effective on many tasks, they often pay attention to regular Euclidean data (e.g., images). However, molecules are usually represented as graphs, which belong to a typical non-Euclidean data. At this time, directly employing the above methods to process graph data does not achieve satisfactory performance. Thus, it is meaningful to design a proper mechanism for learning generalized molecular representation.

**OOD Generalization on Molecular Representation.** Recently, in order to promote the application of biochemical research, learning generalized molecular representation has attracted growing attention. Some methods (Yang et al., 2022; Chen et al., 2022; Fan et al., 2022) follows the "first-separation-then-encoding" paradigm. And they first design a dedicated module to disentangle the original input graph into two different parts. Then, two independent GNN are utilized to encode these two parts, respectively. However, this kind of methods are prone to relying on the separation results. For certain molecules with complex structures, these methods (Yang et al., 2022; Chen et al., 2022; Fan et al., 2022) could not achieve the expected performance. To this end, Zhuang et al. (Zhuang et al., 2023) propose a new strategy, i.e., "first-encoding-then-separation". Concretely, they first employ a GNN to encode molecules. Then, a disentangled module is designed to perform representation decomposition. Though this method has been shown to be effective, for certain biomacromolecules, the proposed score module could not disentangle invariant representations successfully, limiting its application in real-world molecules. In this paper, we explore a new mechanism, i.e., Concept-Enhanced Feedback

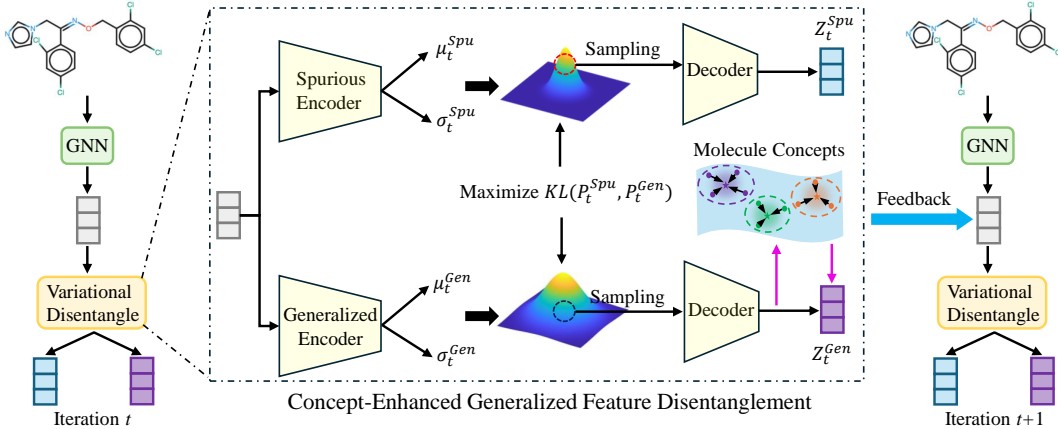

Figure 2: The details of our method for disentangling generalized molecule representation. Our method mainly contains two components: (1) Feedback Disentanglement, and (2) Molecule Concept Extraction. Particularly, based on the current extracted representation, two variational encoders are designed to separately obtain spurious features and generalized features. By maximizing the distance between two estimated distributions, the decomposition ability of two variational encoders could be enhanced. Next, a series of concepts are learned to focus on certain important substructures involving invariant characteristics. By fusing these concepts into the disentangled features, the generalization ability could be further enhanced. Finally, by combining the generalized features from multiple iterations, the output could be taken as the final generalized representation for downstream tasks.

Disentanglement, which aims to utilize the iteration manner for learning generalized molecular representation. Meanwhile, a series of concepts are mined to focus on certain critical substructures, which further enhances the generalization ability. Extensive experimental results on multiple real-world molecule datasets demonstrate that our method could indeed disentangle distribution-agnostic invariant molecular representation.

## 3 CONCEPT-ENHANCED FEEDBACK DISENTANGLEMENT

In this paper, we focus on disentangling generalized molecular representation to reduce the impact of distribution shifts. Here, we first define the OOD generalization problem for MRL.

In general, a molecular graph can be represented as $G = (V, E)$, where $V$ is the graph's node set corresponding to atoms and $E$ denotes the graph's edge sets corresponding to bonds. The goal of MRL task is to predict the target label $\mathcal{Y}$ given the associated input molecule $\mathcal{G}$. The OOD problem on MRL is defined as follows:

$$\min_{f} \max_{e \in \xi} \mathbb{E}_{(\mathcal{G}_i, \mathcal{Y}_i) \backsim P(\mathcal{G}, \mathcal{Y}|e)}[\mathcal{L}(f(\mathcal{G}_i), \mathcal{Y}_i)|e], \tag{1}$$

where $\xi$ represents the support of environments, $f(\cdot)$ is the prediction model and $\mathcal{L}(\cdot, \cdot)$ indicates a loss function. $\mathbb{E}_{(\mathcal{G}_i, \mathcal{Y}_i) \backsim P(\mathcal{G}, \mathcal{Y}|e)}[\mathcal{L}(f(\mathcal{G}_i), \mathcal{Y}_i)|e]$ is called the risk function under a given environment $e$ (Krueger et al., 2021). The reason of resulting in the OOD problem on MRL lies in that the training data only cover very limited environments in $\xi$ while the model is expected to perform well on all environments. Meanwhile, the OOD problem can be categorized into two cases, i.e., covariate and concept shift (Krueger et al., 2021; Widmer & Kubat, 1996). In covariate shift, the distribution of input differs. And the category space of input is the same. Instead, for concept shift, the distribution of input remains the same. But the category space between the training and testing data is inconsistent. In the experiments, we consider these two shift cases.

### 3.1 VARIATIONAL DISENTANGLEMENT FOR MOLECULAR REPRESENTATION

As shown in Fig. 2, our method follows "first-encoding-then-separation" mechanism (Zhuang et al., 2023) for learning generalized molecule representation. Different from this work (Zhuang et al., 2023), we leverage the feedback idea (Zamir et al., 2017) to iteratively extract distribution-agnostic representation, which is more applicable to molecules with highly complex structures. Moreover,

considering that this work (Zhuang et al., 2023) fails to focus on critical molecular substructures, we propose to mine a series of concepts that involve invariant characteristics of certain molecule substructures, which could be used to further improve the generalization of the learned representation. Next, we will introduce the details of our method.

Specifically, given an input molecule $\mathcal{G}$, we first utilize a GNN to encode it. The output is $F$:

$$F = \text{GNN}(\mathcal{G}) \in \mathbb{R}^{|\mathcal{V}| \times d}, \tag{2}$$

where $|\mathcal{V}|$ is the number of atoms in $\mathcal{G}$, and $d$ is the dimensionality of the encoded features. Then, a spurious encoder (i.e., $W_\mu^{Spu}(\cdot)$ and $W_\sigma^{Spu}(\cdot)$) and a generalized encoder (i.e., $W_\mu^{Gen}(\cdot)$ and $W_\sigma^{Gen}(\cdot)$), each consisting of three fully-connected layers, are designed to separately estimate the corresponding distribution. The processes are shown as follows:

$$\mu_t^{Spu} = W_\mu^{Spu}(Q_t), \ \ \sigma_t^{Spu} = W_\sigma^{Spu}(Q_t), \ \ \mu_t^{Gen} = W_\mu^{Gen}(Q_t), \ \ \sigma_t^{Gen} = W_\sigma^{Gen}(Q_t), \tag{3}$$

where $Q_t = \phi([Z_{t-1}^{Gen}, Z_{t-2}^{Gen}])$. $t = 0, \cdots, T$. And $Q_0 = F$, $Q_1 = Z_0^{Gen}$. '$[\cdot, \cdot]$' indicates the concatenation operation. '$\phi(\cdot)$' represents a fully-connected layer that transforms channels. $\mu_t^{Spu} \in \mathbb{R}^{|\mathcal{V}| \times d}$ and $\mu_t^{Gen} \in \mathbb{R}^{|\mathcal{V}| \times d}$ are the estimated means for spurious and generalized features at the current iteration $t$. $\sigma_t^{Spu} \in \mathbb{R}^{|\mathcal{V}| \times d}$ and $\sigma_t^{Gen} \in \mathbb{R}^{|\mathcal{V}| \times d}$ represent the estimated variances for spurious and generalized features. Next, we separately perform a sampling operation to obtain the disentangled features $Z_t^{Spu} \in \mathbb{R}^{|\mathcal{V}| \times d}$ and $\tilde{Z}_t^{Gen} \in \mathbb{R}^{|\mathcal{V}| \times d}$:

$$Z_t^{Spu} = D^{Spu}(\mu_t^{Spu} + \epsilon \cdot \exp(\sigma_t^{Spu})), \ \ \tilde{Z}_t^{Gen} = D^{Gen}(\mu_t^{Gen} + \epsilon \cdot \exp(\sigma_t^{Gen})), \tag{4}$$

where $D^{Spu}(\cdot)$ and $D^{Gen}(\cdot)$ are two defined decoders, each consisting of three fully-connected layers. $\epsilon$ denotes Gaussian noise sampled from $\mathcal{N}(0, I)$.

To promote $Z_t^{Spu}$ and $\tilde{Z}_t^{Gen}$ to separately contain rich spurious and distribution-agnostic information, we first enlarge the distance between the two estimated distributions. Meanwhile, to encourage $\tilde{Z}_t^{Gen}$ to involve plentiful invariant characteristics related to the input molecule, we further define a contrast loss. Finally, the loss $\mathcal{L}_{vd}^t$ for variational disentanglement is defined as follows:

$$\mathcal{L}_{vd}^t = -\log \frac{\exp(\text{sim}(\tilde{Z}_t^{Gen}, F)/\tau)}{\exp(\text{sim}(\tilde{Z}_t^{Gen}, F)/\tau) + \exp(\text{sim}(Z_t^{Spu}, F)/\tau)} - \lambda \cdot KL(P_t^{Spu}, P_t^{Gen}), \tag{5}$$

where $\text{sim}(\tilde{Z}_t^{Gen}, F)$ denotes the average of the cosine similarity between all corresponding elements of $\tilde{Z}_t^{Gen}$ and $F$. $\tau$ and $\lambda$ are two hyper-parameters, which are separately set to 1.0 and 0.01 in the experiments. $P_t^{Spu} = \mathcal{N}(\mu_t^{Spu}, \sigma_t^{Spu})$ and $P_t^{Gen} = \mathcal{N}(\mu_t^{Gen}, \sigma_t^{Gen})$. Obviously, by minimizing the loss $\mathcal{L}_{vd}^t$, the gap between $\tilde{Z}_t^{Gen}$ and $Z_t^{Spu}$ could be enlarged from both semantic and distribution perspectives, which enhances the disentangled ability of our method.

## 3.2 LEARNING MOLECULE CONCEPTS

In general, the critical substructures involving invariant characteristics usually play an important role in downstream tasks. To this end, we explore mining a series of concepts that do not specialize to one particular type or class of molecule, which is instrumental in strengthening the generalization of the learned representation (Locatello et al., 2020).

Concretely, the concepts are defined as $C \curvearrowright \mathcal{N}(\mu_c, \sigma_c) \in \mathbb{R}^{K \times d}$, where $\mu_c$ and $\sigma_c$ are learnable parameters. $K$ indicates the number of concepts. Next, based on the disentangled representation $\tilde{Z}_t^{Gen} \in \mathbb{R}^{|\mathcal{V}| \times d}$, we calculate descriptors that represent molecule-level information:

$$\mathcal{M} = \text{KAN}(\tilde{Z}_t^{Gen}), \ \ \mathcal{H}_i = \sum_{j=1}^{|\mathcal{V}|} \frac{\exp(\mathcal{M}_{j,i})}{\sum_{i=1}^{K} \exp(\mathcal{M}_{j,i})} (\tilde{Z}_{t,j}^{Gen} - C_i), \tag{6}$$

where $i = 1, \cdots, K$. Here, we employ one KAN layer (Liu et al., 2024) to transform $\tilde{Z}_t^{Gen}$, which outputs $\mathcal{M} \in \mathbb{R}^{|\mathcal{V}| \times K}$. Different from MLPs having fixed activation functions on neurons, KANs have learnable activation functions (Liu et al., 2024), which is beneficial for improving the flexibility of the learned concepts. $\mathcal{H} \in \mathbb{R}^{K \times d}$ represents the output descriptors. '$\tilde{Z}_{t,j}^{Gen} - C_i$' denotes the

residual operation. By means of this operation, we can assign molecule substructure features to the corresponding concept $C_i \in \mathbb{R}^d$. Finally, the learned concepts are utilized to enhance the current representation $\tilde{Z}_t^{Gen}$. The corresponding processes are shown as follows:

$$\tilde{\mathcal{H}} = \text{KAN}(\mathcal{H}_r), \quad Z_t^{Gen} = \text{KAN}([\tilde{\mathcal{H}}, \tilde{Z}_t^{Gen}]), \tag{7}$$

where $H_r \in \mathbb{R}^{1 \times Kd}$ is the reshaped result of $\mathcal{H}$. $\tilde{\mathcal{H}} \in \mathbb{R}^{1 \times d}$. $Z_t^{Gen} \in \mathbb{R}^{|\mathcal{V}| \times d}$ is the output generalized representation at the step $t$. By the concatenation operation, the learned concepts could be fused into the current decomposed representation, which is conducive to enhancing the generalization of the output representation $Z_t^{Gen}$. In the experiments, $Z_t^{Gen}$ is utilized as the representation for computing the corresponding task loss, e.g., molecule property prediction, which promotes the learned concepts to capture plentiful task-specific invariant molecule characteristics.

### 3.3 FEEDBACK SEPARATING GENERALIZED MOLECULAR REPRESENTATION

Existing methods (Zhuang et al., 2023; Yang et al., 2022) for learning generalized molecular representation generally employ a one-step disentangled strategy, i.e., directly separating the input into generalized and spurious parts. However, in practice, we may encounter some unknown biomacromolecules containing more atoms and highly complex structures. At this time, using the original one-step mechanism could not obtain satisfactory disentangled results. To this end, we exploit the feedback idea (Zamir et al., 2017) to iteratively separate generalized molecular representation.

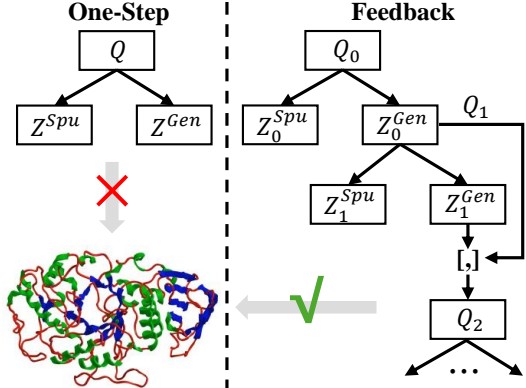

Figure 3: Computation graph of One-Step vs Feedback Disentanglement. For the molecules with complex structures, using one-step disentanglement could not obtain the expected separation results. Instead, the feedback mechanism with multiple iteration steps is beneficial for decomposing complex structures.

As shown in Fig. 3, we assume that the number of feedback iterations is $T$. Given the current feature $Q_t \in \mathbb{R}^{|\mathcal{V}| \times d}$ that is the concatenation of the previous two iteration outputs, we first leverage the above operations to disentangle $Q_t$ into $Z_t^{Spu}$ and $Z_t^{Gen}$, where $t = 0, \cdots, T$. Then, $Q_{t+1} = \phi([Z_{t-1}^{Gen}, Z_t^{Gen}])$ is taken as the input of the current step to repeat the above disentangled operations. In the computation graph of feedback disentanglement (as shown in Fig. 3), we can observe that compared with One-Step disentanglement, the feedback strategy follows a hierarchical disentangled mechanism, which is instrumental in improving the ability of decomposing the molecules with complex structures. Finally, through multiple feedback iterations, we can obtain a series of generalized outputs, i.e., $(Z_0^{Gen}, Z_1^{Gen}, \cdots, Z_T^{Gen})$, and spurious outputs, i.e., $(Z_0^{Spu}, Z_1^{Spu}, \cdots, Z_T^{Spu})$. A self-supervised objective is constructed to close the distance between the output generalized features at each step and enlarge the distance between the decomposed generalized and spurious features at the same step, which strengthens the disentangled ability effectively:

$$\mathcal{L}_{fd} = -\log\left(\frac{1}{T \times T} \sum_{i=0}^{T} \sum_{j=0}^{T} \text{sim}(Z_i^{Gen}, Z_j^{Gen})_{i \neq j} - \beta \cdot \frac{1}{T} \sum_{i=0}^{T} \text{sim}(Z_i^{Gen}, Z_i^{Spu})\right), \tag{8}$$

where $\beta$ is a hyper-parameter, which is set to 0.5 in the experiments. Finally, we take the concatenation result, i.e., $Z^{Gen} = \psi([Z_0^{Gen}, Z_1^{Gen}, \cdots, Z_T^{Gen}]) \in \mathbb{R}^{|\mathcal{V}| \times d}$, as the last generalized representation for downstream tasks, where $\psi(\cdot)$ indicates a fully-connected layer that transforms the channels.

For specific downstream tasks, their goals are to provide the generalized representation $Z^{Gen}$ with predictive capabilities. Since our method is to learn generalized molecule representations, it could be applied to different molecular tasks. During training, the choice of prediction loss function depends on the type of task. For classification loss, we employ the cross-entropy loss, while for regression tasks, we utilize the mean squared error loss. Taking the binary classification task as an example, the cross-entropy loss is computed between the predicted probability $\hat{\mathcal{Y}} = Prob(Z^{Gen})$ and the

Table 1: Evaluation performance on GOOD benchmark. - denotes abnormal results caused by under-fitting declared in the leaderboard, and / denotes that the method cannot be applied to this dataset. For different distribution shifts, our method obtains superior generalization performance.

| Method | GOOD-HIV ↑ | | | | GOOD-ZINC ↓ | | | | GOOD-PCBA ↑ | | | |
| --- | --- | --- | --- | --- | --- | --- | --- | --- | --- | --- | --- | --- |
| | scaffold | | size | | scaffold | | size | | scaffold | | size | |
| | covariate | concept | covariate | concept | covariate | concept | covariate | concept | covariate | concept | covariate | concept |
| ERM | 69.55 | 72.48 | 59.19 | 61.91 | 0.1802 | 0.1301 | 0.2319 | 0.1325 | 17.11 | 21.93 | 17.75 | 15.60 |
| IRM | 70.17 | 71.78 | 59.94 | -(-) | 0.2164 | 0.1339 | 0.6984 | 0.1336 | 16.89 | 22.37 | 17.68 | 15.82 |
| VREx | 69.34 | 72.21 | 58.49 | 61.21 | 0.1815 | 0.1287 | 0.2270 | 0.1311 | 17.10 | 21.65 | 17.80 | 15.85 |
| GroupDRO | 68.15 | 71.48 | 57.75 | 59.77 | 0.1870 | 0.1323 | 0.2377 | 0.1333 | 16.55 | 21.91 | 16.74 | 15.21 |
| Coral | 70.69 | 72.96 | 59.39 | 60.29 | 0.1769 | 0.1303 | 0.2292 | 0.1261 | 17.00 | 22.00 | 17.83 | 16.88 |
| DANN | 69.43 | 71.70 | 62.38 | 65.15 | 0.1746 | 0.1269 | 0.2326 | 0.1348 | 17.20 | 22.03 | 17.71 | 15.78 |
| Mixup | 70.65 | 71.89 | 59.11 | 62.80 | 0.2066 | 0.1391 | 0.2531 | 0.1547 | 16.52 | 20.52 | 17.42 | 13.71 |
| DIR | 68.44 | 71.40 | 57.67 | 74.39 | 0.3682 | 0.2543 | 0.4578 | 0.3146 | 16.33 | 23.82 | 16.04 | 16.80 |
| GSAT | 70.07 | 72.51 | 60.73 | 56.96 | 0.1418 | 0.1066 | 0.2101 | 0.1038 | 16.45 | 20.18 | 17.57 | 13.52 |
| GREA | 71.98 | 70.76 | 60.11 | 60.96 | 0.1691 | 0.1157 | 0.2100 | 0.1273 | 16.28 | 20.23 | 17.12 | 13.82 |
| CAL | 69.12 | 72.49 | 59.34 | 56.16 | / | / | / | / | 15.87 | 18.62 | 16.92 | 13.01 |
| DisC | 58.85 | 64.82 | 49.33 | 74.11 | / | / | / | / | / | / | / | / |
| MoleOOD | 69.39 | 69.08 | 58.63 | 55.90 | 0.2752 | 0.1996 | 0.3468 | 0.2275 | 12.90 | 12.92 | 12.64 | 10.30 |
| CIGA | 69.40 | 71.65 | 61.81 | 73.62 | / | / | / | / | / | / | / | / |
| iMoLD | 72.93 | 74.32 | 62.86 | 77.43 | 0.1410 | 0.1014 | 0.1863 | 0.1029 | 17.32 | 22.58 | 18.02 | 18.21 |
| **Ours (CFD)** | **76.42** | **77.83** | **64.14** | **79.28** | **0.1187** | **0.0765** | **0.1421** | **0.0852** | **19.78** | **25.64** | **19.18** | **20.03** |

ground-truth label $\mathcal{Y}$:

$$\mathcal{L}_{pred} = \mathcal{Y} \log \hat{\mathcal{y}} + (1 - \mathcal{Y}) \log(1 - \hat{\mathcal{y}}), \tag{9}$$

where $Prob(\cdot)$ represents a function that computes probability. Finally, the learning objective can be defined as the weighted sum of the above losses:

$$\mathcal{L}_{\text{task}} = \mathcal{L}_{pred} + \alpha_1 \cdot \mathcal{L}_{fd} + \alpha_2 \cdot \sum_{t=0}^{T} \mathcal{L}_{vd}^t, \tag{10}$$

where $\alpha_1$ and $\alpha_2$ are hyper-parameters, which are separately set to 0.1 and 0.01 in the experiments.

## 4 EXPERIMENTS

In this section, we conduct extensive experiments to answer the research questions. (1) Can our method CFD achieve better OOD generalization performance against baselines? (2) Does our method possess the ability to capture important substructures and improve the performance of molecular substructure prediction? (3) How does each component contribute to the final performance?

### 4.1 EXPERIMENTAL SETUP

**Datasets.** For OOD molecular representation learning, we follow the settings of the work (Zhuang et al., 2023) and employ two real-world datasets, i.e., GOOD (Gui et al., 2022) that is a systematic benchmark tailored specifically for graph OOD problems, and DrugOOD (Ji et al., 2022) that is a OOD benchmark for AI-aided drug discovery. Besides, for molecule's ground-state prediction, we follow the settings of the work (Xu et al., 2024) and utilize Molecule3D (Xu et al., 2021b) and QM9 (Ramakrishnan et al., 2014) to evaluate the ability of our method for focusing on substructures. In Appendix, we will introduce more details about these datasets.

**Implementation Details.** Our method is used for learning specific molecule representation and could be plugged into existing methods (Zhuang et al., 2023; Xu et al., 2024). The number of the learned concepts is set to 12. The number of feedback iterations is set to 8. It is worth noting that during feedback, we do not introduce new parameters.

**Baselines.** For OOD molecular representation learning, we mainly compare our method with some common approaches for non-Euclidean data, i.e., ERM (Vapnik, 2013), IRM (Arjovsky et al., 2019), VREx (Krueger et al., 2021), GroupDRO (Sagawa et al., 2019), Coral (Sun & Saenko, 2016), DANN (Ganin et al., 2016), Mixup (Zhang, 2017), and iMoLD (Zhuang et al., 2023). Meanwhile, our method is compared with some graph-specific algorithms, e.g., graph OOD algorithms such as CAL (Sui et al., 2022), DisC (Fan et al., 2022), MoleOOD (Yang et al., 2022), and CIGA (Chen et al., 2022), as well as interpretable graph learning methods, e.g., DIR (Wu et al., 2022), GSAT (Miao et al., 2022), and GREA (Liu et al., 2022).

## 4.2 ANALYSIS OF OOD MOLECULAR REPRESENTATION LEARNING

To evaluate the generalization ability of our method, our method is evaluated on GOOD benchmark (Gui et al., 2022). Table 1 shows the performance. We can observe that some graph-specific OOD methods could achieve effective performance on some synthetic datasets (e.g., to predict whether a specific motif is present in a synthetic graph). However, they do not perform well on molecules with complex data structures and distributions, which shows that our method could apply to molecules with complex structures and improve the generalization performance. Besides, the performance of iMoLD (Zhuang et al., 2023) outperforms MoleOOD (Yang et al., 2022), which shows the effectiveness of the strategy "first-encoding-then-separation". However, iMoLD (Zhuang et al., 2023) fails to capture the critical substructures that involve certain invariant characteristics, which may affect its performance. We can see that our method improves the performance significantly. This indicates that the feedback disentanglement could indeed decompose generalized features. Meanwhile, our method could be applied to complex molecules. This also shows that the learned concepts are beneficial for capturing important substructures, which further strengthens the generalization ability.

Table 2 further evaluates our method on DrugOOD dataset (Ji et al., 2022). Particularly, the recent work MILI (Wang et al., 2024a) follows "first-separation-then-encoding" mechanism. And it first identifies privileged substructure and develops a dual-head GNN to achieve invariant representations of substructures. When the molecular substructures are accurately identified, the generalization performance of OOD molecule representation could be improved effectively. We can see that our method still outperforms iMoLD (Zhuang et al., 2023) and MILI (Wang et al., 2024a). This demonstrates that using the feedback iteration idea is beneficial for extracting generalized molecular characteristics. By fusing the concepts capturing plentiful substructure content, the generalization ability could be further enhanced.

Table 2: Evaluation performance on DrugOOD dataset.

| Method | IC50 ↑ | | | EC50 ↑ | | |
|---|---|---|---|---|---|---|
| | Assay | Scaffold | Size | Assay | Scaffold | Size |
| ERM | 71.63 | 68.79 | 67.50 | 67.39 | 64.98 | 65.10 |
| IRM | 71.15 | 67.22 | 61.58 | 67.77 | 63.86 | 59.19 |
| Coral | 71.28 | 68.36 | 64.53 | 72.08 | 64.83 | 58.47 |
| MixUp | 71.49 | 68.59 | 67.79 | 67.81 | 65.77 | 65.77 |
| DIR | 69.84 | 66.33 | 62.92 | 65.81 | 63.76 | 61.56 |
| GSAT | 70.59 | 66.45 | 66.70 | 73.82 | 64.25 | 62.65 |
| GREA | 70.23 | 67.02 | 66.59 | 74.17 | 64.50 | 62.81 |
| CAL | 70.09 | 65.90 | 66.42 | 74.54 | 65.19 | 61.21 |
| DisC | 61.40 | 62.70 | 61.43 | 63.71 | 60.57 | 57.38 |
| MoleOOD | 71.62 | 68.58 | 65.62 | 72.69 | 65.74 | 65.51 |
| CIGA | 71.86 | 69.14 | 66.92 | 69.15 | 67.32 | 65.65 |
| iMoLD | 72.11 | 68.84 | 67.92 | 77.48 | 67.79 | 67.09 |
| MILI | 72.67 | 69.58 | 68.40 | 77.11 | 68.07 | 65.97 |
| **Ours (CFD)** | **73.86** | **70.02** | **69.73** | **78.32** | **69.13** | **67.62** |

Table 3: Performance of molecule's ground-state prediction based on Molecule3D and QM9 datasets.

| | Validation | | | Test | | |
|---|---|---|---|---|---|---|
| | D-MAE↓ | D-RMSE↓ | C-RMSD↓ | D-MAE↓ | D-RMSE↓ | C-RMSD↓ |
| (a) Molecule3D Random Split | | | | | | |
| RDKit DG | 0.581 | 0.930 | 1.054 | 0.582 | 0.932 | 1.055 |
| RDKit ETKDG | 0.575 | 0.941 | 0.998 | 0.576 | 0.942 | 0.999 |
| DeeperGCN-DAGNN (Xu et al., 2021b) | 0.509 | 0.849 | - | 0.571 | 0.961 | - |
| GINE (Hu et al., 2019) | 0.590 | 1.014 | 1.116 | 0.592 | 1.018 | 1.116 |
| GATv2 (Brody et al., 2021) | 0.563 | 0.983 | 1.082 | 0.564 | 0.986 | 1.083 |
| GPS (Rampášek et al., 2022) | 0.528 | 0.909 | 1.036 | 0.529 | 0.911 | 1.038 |
| GTMGC (Xu et al., 2024) | 0.432 | 0.719 | 0.712 | 0.433 | 0.721 | 0.713 |
| **GTMGC + Ours** | **0.397** | **0.682** | **0.684** | **0.407** | **0.695** | **0.688** |
| (b) QM9 | | | | | | |
| RDKit DG | 0.358 | 0.616 | 0.722 | 0.358 | 0.615 | 0.722 |
| RDKit ETKDG | 0.355 | 0.621 | 0.691 | 0.355 | 0.621 | 0.689 |
| GINE (Hu et al., 2019) | 0.357 | 0.673 | 0.685 | 0.357 | 0.669 | 0.693 |
| GATv2 (Brody et al., 2021) | 0.339 | 0.663 | 0.661 | 0.339 | 0.659 | 0.666 |
| GPS (Rampášek et al., 2022) | 0.326 | 0.644 | 0.662 | 0.326 | 0.640 | 0.666 |
| GTMGC (Xu et al., 2024) | 0.262 | 0.468 | 0.362 | 0.264 | 0.470 | 0.367 |
| **GTMGC + Ours** | **0.223** | **0.434** | **0.305** | **0.218** | **0.442** | **0.309** |

## 4.3 ANALYSIS OF MOLECULE'S GROUND-STATE PREDICTION

The molecular ground-state conformation refers to the lowest energy state of a molecule on its potential energy surface (Xu et al., 2024). It belongs to substructures of original molecules, which

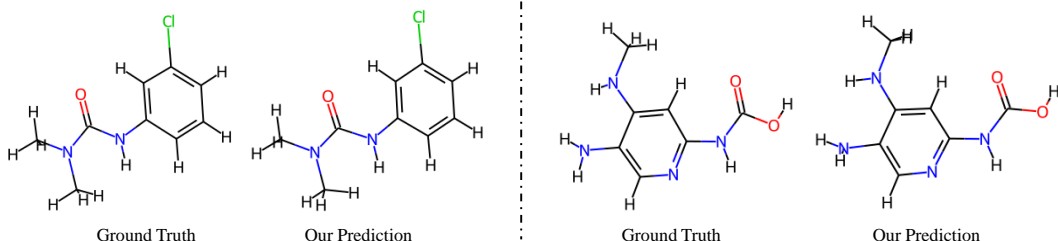

Figure 4: The prediction results of the molecule's ground-state conformation. Compared with ground truth, our method could localize the critical substructures, improving the prediction accuracy.

play a crucial role in determining the physical, chemical, and biological properties of molecules. By evaluating this task, we can directly observe CFD's ability for capturing critical substructures. Here, our method is directly plugged into GTMGC (Xu et al., 2024) that is a transformer architecture. The training processes and optimization objectives are kept unchanged. In Table 3, we can observe that plugging our method could improve GTMGC's prediction performance effectively. This indicates that our method is able to capture critical substructure characteristics. Meanwhile, our method could be plugged into Transformer architecture, showing our method's flexibility.

In Fig. 4, we provide two prediction examples from our method. We can observe that compared with ground truth results, our method could accurately capture those critical substructures, further demonstrating the effectiveness of our method.

Table 4: Analysis of our CFD method. 'Feedback' and 'Concepts' separately indicate feedback disentanglement and molecule concept learning. 'Cov-Sold', 'Cet-Sold', 'Cov-Size', and 'Cet-Size' denote scaffold-covariate, scaffold-concept, size-covariate, and size-concept shifts, respectively.

| Feedback | Concepts | Cov-Sold | Cet-Sold | Cov-Size | Cet-Size |
|:---:|:---:|:---:|:---:|:---:|:---:|
| ✓ | | 0.1391 | 0.0923 | 0.1754 | 0.0983 |
| | ✓ | 0.1685 | 0.1172 | 0.2046 | 0.1268 |
| ✓ | ✓ | **0.1187** | **0.0765** | **0.1421** | **0.0852** |

## 4.4 ABLATION ANALYSIS

To evaluate the effectiveness of each component, we take GOOD-ZINC (Gui et al., 2022) as the dataset to make an ablation analysis.

**Analysis of our method.** As shown in Fig. 2, our method mainly consists of two components, i.e., feedback disentanglement and molecule concept mining. Meanwhile, these two modules could be taken as the independent module. In Table 4, we make an ablation experiment. 'Concepts' represents that the concept learning module is plugged into ERM (Vapnik,

Table 5: Analysis of feedback iteration number.

| Iterations | Cov-Sold | Cet-Sold | Cov-Size | Cet-Size |
|:---:|:---:|:---:|:---:|:---:|
| 1 | 0.1378 | 0.1012 | 0.1789 | 0.1021 |
| 4 | 0.1285 | 0.1008 | 0.1573 | 0.0963 |
| 8 | 0.1187 | **0.0765** | **0.1421** | 0.0852 |
| 12 | 0.1191 | 0.0799 | 0.1453 | **0.0844** |
| 16 | **0.1172** | 0.0805 | 0.1479 | 0.0867 |

2013). We can see that for different shifts, using feedback disentanglement and learning molecule concepts could improve the performance significantly. This shows that our method could effectively separate generalized features from spurious features. Meanwhile, the mined concepts could capture critical substructure characteristics, further enhancing the disentangled features' generalization.

**Number of feedback iterations.** To achieve the expected generalized representation, we employ multiple feedback iterations. In Table 5, we can see that the iteration number affects the generalization performance. For molecules with complex structures, using fewer iterations could not decompose the expected generalized representation effectively, which weakens the

Table 6: Analysis of molecule concept number. Here, we only change the concept number. Other operations and settings are kept unchanged.

| Number | Cov-Sold | Cet-Sold | Cov-Size | Cet-Size |
|:---:|:---:|:---:|:---:|:---:|
| 8 | 0.1233 | 0.0892 | 0.1541 | 0.0927 |
| 12 | **0.1187** | 0.0765 | **0.1421** | **0.0852** |
| 16 | 0.1201 | **0.0743** | 0.1478 | 0.0874 |

performance. Meanwhile, we observe that the performance change is very small when the iteration number is larger than 12 for the given dataset. For our method, the performance of using 8 feedback iterations is the best.

**Number of molecule concepts.** As shown in Fig. 2, we explore extracting a series of molecule concepts to focus on critical substructures. In Table 6, we can observe that the concept number affects the performance. Utilizing fewer concepts could not capture sufficient substructure characteristics

effectively, which reduces the performance. For our method, when the concept number is set to 12, the generalization performance is the best. Besides, during mining molecule concepts (equation 6 and equation 7), we employ KAN (Liu et al., 2024) to improve the concept accuracy. Here, we find that replacing KAN with MLP results in performance degradation. For example, for Cov-Sold case, the performance is decreased by around 0.6%, indicating KAN's effectiveness.

## 5 CONCLUSION

In this paper, we propose a new method, i.e., Concept-Enhanced Feedback Disentanglement, for learning generalized molecular representation against distribution shifts. Particularly, by performing multiple feedback iterations, our method could progressively decompose expected features involving rich generalized information. Meanwhile, fusing the concepts that focus on substructures could further strengthen the generalization. Extensive experimental results on multiple datasets and tasks demonstrate that our method could indeed disentangle generalized molecular representations.

## ACKNOWLEDGEMENT

This work is supported in part by the National Key R&D Program of China (No. 2023YFC3305600), Joint Fund of Ministry of Education of China (8091B022149, 8091B02072404), National Natural Science Foundation of China (62472333, 62132016, 62171343), and Natural Science Basic Research Program of Shaanxi (2020JC-23).

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

## A APPENDIX

To learn generalized molecular representation, this paper proposes a new method, i.e., Concept-Enhanced Feedback Disentanglement, aiming to use feedback decomposition and concepts involving critical substructures to improve the generalization of the learned representation. In the appendix, we provide more experimental details, additional analyses, and more prediction results.

## B DATASETS AND METRICS

**Datasets.** For molecular representation learning, we employ two benchmarks, i.e., GOOD (Gui et al., 2022) and DrugOOD (Ji et al., 2022). Particularly, we follow the work (Zhuang et al., 2023) to utilize three molecular datasets for the prediction task: (1) GOOD-HIV (Wu et al., 2018), where the purpose is binary classification to predict whether a molecule can inhibit HIV; (2) GOOD-ZINC (Gómez-Bombarelli et al., 2018), which is a regression dataset aimed at predicting molecular solubility; and (3) GOOD-PCBA (Wu et al., 2018), which includes 128 bioassays and forms 128 binary classification tasks. Each dataset contains two environment-splitting strategies (scaffold and size), and two shift types (covariate and concept) are applied per splitting outcome, resulting in a total of 12 distinct datasets (as shown in Table 1). Furthermore, DrugOOD (Ji et al., 2022) provides three environment-splitting strategies, including assay, scaffold, and size, and applies these three splitting to two measurements (IC50 and EC50). As a result, we obtain 6 datasets (as shown in Table 2), and each dataset contains a binary classification task for drug target binding affinity prediction

For molecule's ground-state prediction, we follow the work (Xu et al., 2024) to utilize Molecule3D (Xu et al., 2021b) and QM9 (Wu et al., 2018) as benchmarks. Particularly, Molecule3D aims to use DNNs to predict the ground-state 3D geometries of molecules based solely on their 2D graph structure. And this large-scale dataset comprises approximately 4 million molecules, each with its own 2D molecular graph, ground-state 3D geometric structure, and four additional quantum properties. In this paper, we employ the random splitting according to the same distribution based on the molecule's core component. Besides, QM9 is a quantum chemistry dataset that provides geometry, energy, electronic, and thermodynamic properties for nearly 130,000 organic molecules with 9 heavy atoms, containing molecular most stable conformation.

**Metrics.** For molecular representation learning, we report the ROC-AUC score for GOOD-HIV and DrugOOD datasets as the task is binary classification. For GOOD-ZINC, we use the Mean Average Error (MAE) since the task is regression. While for GOOD-PCBA, we use Average Precision (AP) averaged over all tasks as the evaluation metric due to extremely imbalanced classes.

For molecule's ground-state prediction, we follow the work (Xu et al., 2024) to utilize three metrics, i.e., D-MAE, D-RMSE, and C-RMSD. Concretely, given a dataset with $N$ interatomic distances, the Mean Absolute Error (MAE) and Root Mean Square Error (RMSE) between the prediction $\{d_i^*\}_{i=1}^N$ and the ground truth $\{d_i\}_{i=1}^N$ are used to evaluate the performance at node-pair level:

$$\mathrm{D-MAE}(\{d_i\}_{i=1}^N, \{d_i^*\}_{i=1}^N) = \frac{1}{N}\sum_{i=1}^N |d_i - d_i^*|, \tag{11}$$

$$\mathrm{D-RMSE}(\{d_i\}_{i=1}^N, \{d_i^*\}_{i=1}^N) = \sqrt{\frac{1}{N}\sum_{i=1}^N (d_i - d_i^*)^2}. \tag{12}$$

Additionally, C-RMSD between the ground truth $\mathbf{G}$ and the prediction $\mathbf{G}^*$ is computed as follows:

$$\mathrm{C-RMSD}(\mathbf{G}, \mathbf{G}^*) = \sqrt{\frac{1}{n}\sum_{i=1}^n \|g_i - g_i^*\|_2^2}. \tag{13}$$

## C THE CAUSE OF MOLECULAR DISTRIBUTION SHIFT

For molecule data, there exist many kinds of distribution shifts. In this paper, we investigate three types of shift strategies, i.e., scaffold, size, and assay. Here, we give more explanations.

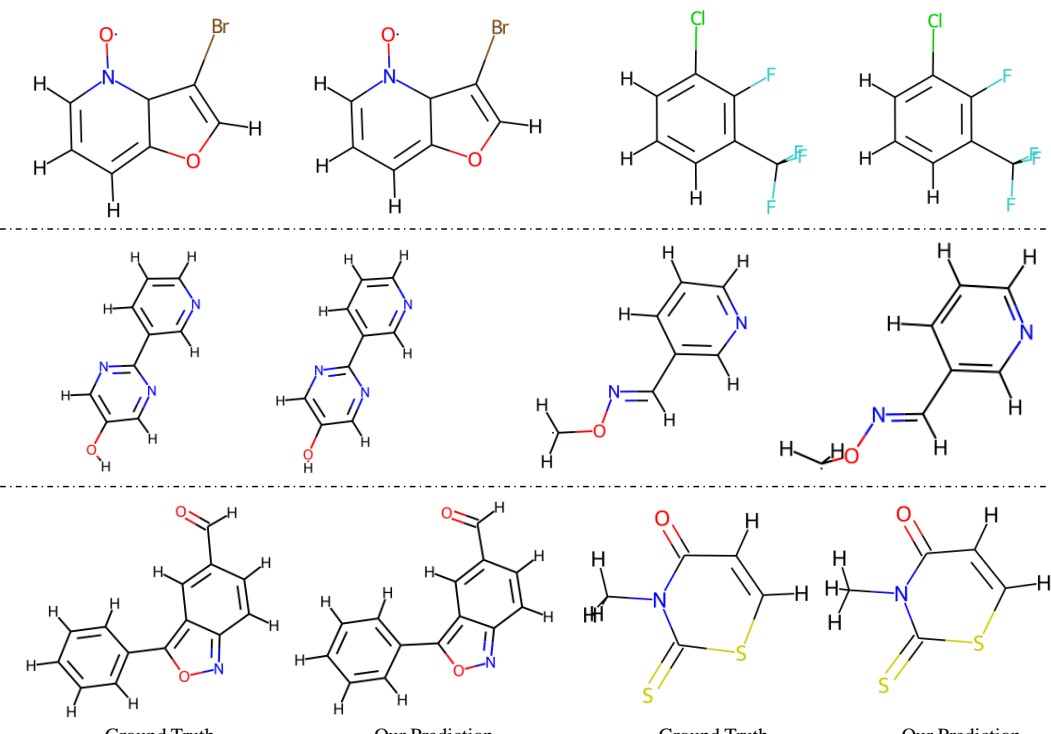

Figure 5: More prediction results of molecule's ground-state conformation. We can observe that our method could accurately localize critical ground-state parts that determine the physical, chemical, and biological properties of the molecule in most cases, which demonstrates the effectiveness of our method for learning generalized molecule representation and capturing substructure characteristics.

(1) **Scaffold Environment**: molecular scaffold is the fundamental structure of a molecule with desirable bioactive properties. Molecules with the same scaffold belong to the same environment. Distribution shift arises when there is a change in the molecular scaffold.

(2) **Size Environment**: the size of a molecule refers to the total number of atoms in the molecule. Molecular size is also an inherent structural characteristics of molecular graphs. The distribution shift occurs when size changes.

(3) **Assay Environment**: assay is an experimental method as an examination or determination for molecular characteristics. Due to variations in assay environments and targets, activity values measured by different assays often differ significantly. Samples tested within the same assay belong to a single environment, while a change in the assay leads to a distribution shift.

## D   MORE ABLATION EXPERIMENTS OF HYPER-PARAMETERS

In equation 10, we utilize two hyper-parameters, i.e., $\alpha_1$ and $\alpha_2$, to balance the two loss terms, i.e., $\mathcal{L}_{fd}$ and $\mathcal{L}_{vd}$. Here, based on GOOD-ZINC with Covariate-Scaffold shift, we make an ablation analysis. We only change the settings of hyper-parameters.

**Analysis of hyper-parameter** $\alpha_1$. In equation 10, the goal of the loss term $\mathcal{L}_{fd}$ is to encourage the feedback disentangled process to decompose generalized representation effectively. Thus, the setting of hyper-parameter $\alpha_1$ is important. Here, we make an experiment. When $\alpha_1$ is separately set to 0.5, 0.1, and 0.05, the corresponding performance is 0.1251, **0.1187**, and 0.1239.

**Analysis of hyper-parameter** $\alpha_2$. In equation 10, the loss term $\mathcal{L}_{vd}^t$ is to promote the decomposed features to involve plentiful invariant characteristics related to the input molecule while enlarging the estimated distribution gap between the generalized and spurious features. Thus, the hyper-parameter $\alpha_2$ is still an important hyper-parameter. Here, we make an experiment. When $\alpha_2$ is separately set to 0.05, 0.01, and 0.005, the performance is 0.1248, **0.1187**, and 0.1225.

---

**Algorithm 1** Training Process of Concept-Enhanced Feedback Disentanglement

---

**Input:** Molecule Dataset $\{G, Y\}$, randomly initialized GNN, randomly initialized spurious encoder and generalized encoder, randomly initialized concept learning module, weight $\alpha_1$ for the loss $\mathcal{L}_{fd}$, weight $\alpha_2$ for the loss $\mathcal{L}_{vd}^t$.

**Output:** Trained parameters.

**while** *train* **do**

    Sample molecules from the given dataset $\{G, Y\}$.

    Utilize a GNN to extract the representation $F$.

    **for** $t = 0$ **do**

        $Q_0 = F$.

        Use equation 3 to estimate corresponding means (i.e., $\mu_0^{Spu}$ and $\mu_0^{Gen}$) and variances (i.e., $\sigma_0^{Spu}$ and $\sigma_0^{Gen}$) for spurious and generalized features.

        Perform sampling and encoding operations to obtain $Z_0^{Spu}$ and $\tilde{Z}_0^{Gen}$ based on equation 4.

        Learn molecule concepts $C$ and calculate the generalized representation $Z_0^{Gen}$ based on equation 6 and equation 7.

    **end**

    **for** $t = 1$ **do**

        $Q_1 = Z_0^{Gen}$.

        Use equation 3 to estimate corresponding means (i.e., $\mu_1^{Spu}$ and $\mu_1^{Gen}$) and variances (i.e., $\sigma_1^{Spu}$ and $\sigma_1^{Gen}$) for spurious and generalized features.

        Perform sampling and encoding operations to obtain $Z_1^{Spu}$ and $\tilde{Z}_1^{Gen}$ based on equation 4.

        Learn molecule concepts $C$ and calculate the generalized representation $Z_1^{Gen}$ based on equation 6 and equation 7.

    **end**

    **for** $t = 2, ..., T$ **do**

        $Q_t = \phi([Z_{t-1}^{Gen}, Z_{t-2}^{Gen}])$.

        Use equation 3 to estimate corresponding means (i.e., $\mu_t^{Spu}$ and $\mu_t^{Gen}$) and variances (i.e., $\sigma_t^{Spu}$ and $\sigma_t^{Gen}$) for spurious and generalized features.

        Perform sampling and encoding operations to obtain $Z_t^{Spu}$ and $\tilde{Z}_t^{Gen}$ based on equation 4.

        Learn molecule concepts $C$ and calculate the generalized representation $Z_t^{Gen}$ based on equation 6 and equation 7.

    **end**

    Calculate the overall training objective $\mathcal{L}_{\text{task}}$ using equation 5, equation 8, equation 9.

    Update the trained parameters based on equation 10.

**end**

---

Finally, Algorithm 1 shows the training processes of our method. For step $t >= 2$, the feedback is from the outputs of the previous two steps. By means of feedback disentanglement, our method could decompose generalized molecular representation effectively, which improves the performance of downstream tasks. In Fig. 5, we show more prediction results of molecule's ground-state conformation. We can see that our method could localize critical ground-state molecular substructures, which further demonstrates the superiorities of our method.

## E MORE CHEMICAL INSPIRATION OF OUR METHOD

This method is mainly based on the following chemical observations:

(1) A chemical molecule is a complex composed of multiple atoms and bonds. The reason for the OOD problem is the structural change of molecules. For example, changes in molecular chirality can lead to deviations in molecular properties. Thus, learning robust molecule representation is important for molecule-based tasks, e.g., drug discovery.

(2) Lines 082-095 have indicated that existing methods could be classified into two types, i.e., First-Separation-Then-Encoding and First-Encoding-Then-Separation (as shown in Fig. 1 (a) and (b)). As indicated in Lines 272-283 and Fig. 3, these methods for learning generalized molecular representation generally employ a one-step disentangled strategy, i.e., directly separating the input into

generalized and spurious parts. However, in practice, we may encounter some unknown biomacro-molecules containing more atoms and highly complex structures, e.g., entanglement of multiple substructures. At this time, using the original one-step mechanism could not obtain satisfactory disentangled results. To this end, we exploit the feedback idea to iteratively and progressively separate generalized molecular representation.

(3) In general, the molecule's ground-state conformation belongs to the substructures of molecules and determines their properties. Therefore, it is important to capture substructure-related characteristics. To this end, we define a series of concepts to build connections between molecule substructures and corresponding concepts, which further improves the performance.

## F    MORE EXPERIMENTS TO EVALUATE INVARIANCE

Table 7: To further demonstrate the ability of disentangling invariance, we further evaluate our method on Two-Piece Graph Datasets Chen et al. (2024).

| Datasets | {0.8, 0.9} | {0.7, 0.9} | Avg. |
|---|---|---|---|
| DisC Fan et al. (2022) | 45.06 | 39.42 | 42.24 |
| CIGA Chen et al. (2022) | 57.87 | 43.62 | 50.75 |
| GALA Chen et al. (2024) | 76.42 | 72.50 | 74.46 |
| **Ours** | **78.26** | **74.32** | **76.29** |

We follow the work Chen et al. (2024) and further evaluate our method on Two-Piece Graph Datasets Chen et al. (2024). Particularly, each dataset is generated from a variation of two-piece graph model, denoted as {a, b}, where a refers to the invariant correlation strength and b refers to the spurious correlation strength. The results are shown in Table 7.

We can observe that our method indeed improves the performance. Particularly, when the invariant correlation strength is weaker than the spurious correlation strength, the performance of our method outperforms state-of-the-art methods, further demonstrating that our method is indeed beneficial for disentangling invariant characteristics.

## G    COMPARISON BETWEEN RVQ AND MOLECULE CONCEPT LEARNING

Our molecule concept learning module is significantly different from Residual Vector Quantization (RVQ) Zhuang et al. (2023). In essence, RVQ is to perform feature replacement. Our method is to build the connection to capture critical molecule substructures:

(1) **The motivation is different**. RVQ aims to leverage the discretized continuous representation to improve the model generalization. As shown in Fig. 6, our method is to learn a series of concepts that could capture critical molecule structure information involving plentiful task-specific invariant molecule characteristics.

(2) **The operations are different**. RVQ first introduces a shared learnable codebook as a discrete latent space. For each node representation in molecular graph, RVQ looks up and fetches the nearest neighbor in the codebook and outputs it as the result. Meanwhile, a sum operation is further used to strengthen the representation ability. During training, similar to VQ-VAE Van Den Oord et al. (2017), RVQ employs the exponential moving average updates for the codebook.

Differently, our concepts are first sampled from a learnable Gaussian distribution. And we employ KAN operation to promote the learning of concepts. Importantly, as shown in equation 6, the weighted residual operation is to align critical molecule substructure features to the corresponding concept, building the connection between molecule substructures and corresponding concepts. During training, we only utilize the task loss, e.g., molecule property prediction, to promote the learned concepts to capture plentiful task-specific invariant molecule characteristics.

(3) **Experimental results show the effectiveness of our method.** Taking GOOD-ZINC as the example, for iMoLD Zhuang et al. (2023), we first remove the RVQ operation and keep other operations unchanged. We observe that removing RVQ does not affect the performance. Instead, our concept mining module could be plugged into iMoLD to further improve its performance. For the

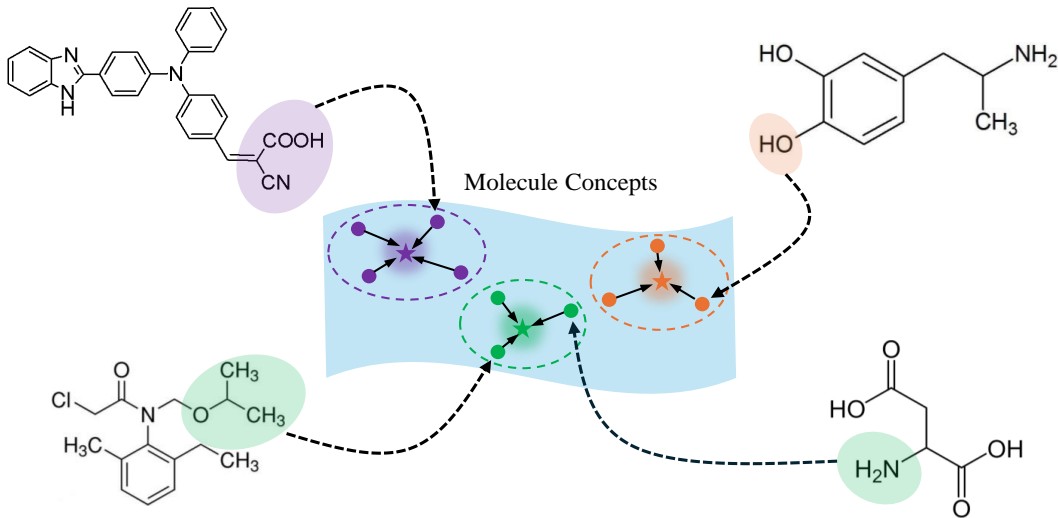

Figure 6: Molecule concepts (colorful stars) are learned from all molecule structures, which are not specific to certain molecule structures. By means of the residual operation (as shown in solid arrows), we can build the connection between molecule substructures and corresponding concepts, which is beneficial for aligning critical molecule substructure features to the corresponding concept.

covariate case of scaffold, the performance is improved by 0.0142. Furthermore, the prediction results of the molecule's ground-state conformation in Fig. 4 and 5 also show that our learned concepts are beneficial for capturing critical substructures.

## H   FURTHER ANALYSIS OF THE LOSS FUNCTION

equation 10 is the optimization objective, consisting of three terms, i.e., $\mathcal{L}_{pred}$, $\mathcal{L}_{fd}$ (as shown in equation 8), and $\mathcal{L}_{vd}^t$ (as shown in equation 5).

**Proposition 1.** $\mathcal{L}_{vd}^t$ mainly is a contrastive loss. Minimizing $\mathcal{L}_{vd}^t$ could enlarge the gap between the disentangled $\tilde{Z}_t^{Gen}$ and $Z_t^{Spu}$. Meanwhile, $\tilde{Z}_t^{Gen}$ is promoted to retain plentiful input-related content.

**Analysis.** The first term of $\mathcal{L}_{vd}^t$ is related to the InfoNCE loss Oord et al. (2018). Concretely, we set $\mathcal{B}_1 = \exp(\text{sim}(\tilde{Z}_t^{Gen}, F))$ and $\mathcal{B}_2 = \exp(\text{sim}(Z_t^{Spu}, F))$:

$$\mathbb{E}[\log\frac{\mathcal{B}_1}{\mathcal{B}_1 + \mathcal{B}_2}] = \mathbb{E}[\log\mathcal{B}_1] - \mathbb{E}[\log(\mathcal{B}_1 + \mathcal{B}_2)] \tag{14}$$
$$\leq \mathbb{E}[\log\mathcal{B}_1] - \mathbb{E}[\log\mathcal{B}_2]$$

Obviously, maximizing equation 14 narrows the semantic distance between $\tilde{Z}_t^{Gen}$ and the input $F$ and enlarges the semantic gap between $Z_t^{Spu}$ and $F$. Equally, this operation enlarges the gap between $\tilde{Z}_t^{Gen}$ and $Z_t^{Spu}$, which is beneficial for strengthening the disentangled ability.

**Proposition 2.** Minimizing $\mathcal{L}_{fd}$ is instrumental in achieving the invariant representation $Z^{Gen}$.

**Analysis.** In equation 8, $\mathcal{L}_{fd}$ mainly consists of two terms, i.e., $\sum_{i=0}^T \sum_{j=0}^T \text{sim}(Z_i^{Gen}, Z_j^{Gen})_{i\neq j}$ and $\sum_{i=0}^T \text{sim}(Z_i^{Gen}, Z_i^{Spu})$:

$$\sum_{i=0}^T \sum_{j=0}^T \text{sim}(Z_i^{Gen}, Z_j^{Gen})_{i\neq j} - \sum_{i=0}^T \text{sim}(Z_i^{Gen}, Z_i^{Spu}) \tag{15}$$

Maximizing equation 15 first enlarges the gap between $Z_i^{Gen}$ and $Z_i^{Spu}$ at the $i$-th iteration, which is beneficial for enhancing the disentangled ability. Secondly, $Z_i^{Gen}$ and $Z_j^{Gen}$ separately represent disentangled generalized outputs at different iterations. Here, we assume that when taking a representation involving plentiful invariant information as the input, the disentangled output should contain less spurious information, i.e., the output is equal to the input. Therefore, narrowing the semantic distance is conducive to learning invariant representations across various environments.

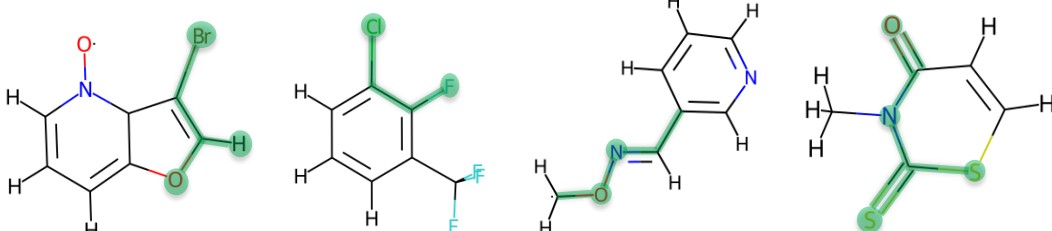

Figure 7: Visualization examples of molecule concepts. Here, we directly project the captured concepts into the predicted molecules. The green regions represent the focused content.

## I EXAMINING IMPORTANCE OF CONCEPTS ACROSS ITERATIONS

Table 5 analyzes the impact of iteration number. We can observe that when the iteration number increases, inserting concepts could improve the performance effectively, which indicates the effectiveness of the learned concepts. To further analyze the impact of iteration number on concepts, we also make a new ablation experiment, i.e., removing the concepts at the 4th, and 8th iteration. The results are shown as follows:

Table 8: Examining importance of concepts across iterations.

| Iterations | Concepts | CovSold | CetSold | CovSize | CetSize |
|---|---|---|---|---|---|
| 4 | ✓ | 0.1285 | 0.1008 | 0.1573 | 0.0963 |
| 4 | – | 0.1402 | 0.1117 | 0.1738 | 0.1102 |
| 8 | ✓ | 0.1187 | 0.0765 | 0.1421 | 0.0852 |
| 8 | – | 0.1189 | 0.0783 | 0.1449 | 0.0897 |

We can observe that when the iteration number is small, inserting concepts could significantly affect the performance. Instead, when the iteration number is large, the impact of concepts is somewhat weak. The reason may be that when the iteration number is small, leveraging concepts is beneficial for capturing critical substructures, improving the performance. As the iteration number increases, concepts progressively pay attention to meaningful substructure characteristics, reducing the iteration impact on performance.

## J MORE ANALYSIS OF MOLECULE CONCEPTS

We perform an ablation experiment of the decoder that calculates task-related representation. We observe that for classification and regression tasks, using decoders with more layers leads to the performance degradation. The reason may be that this operation introduces more parameters, resulting in the overfitting risk.

In Fig. 7, we show some visualization examples of molecule concepts. Since these concepts are learned from all molecule data, they are not specific to certain kinds of molecules, which is beneficial for improving the generalization of molecule concepts. For different predictions, we can observe that the learned concepts could indeed focus on critical molecule substructures, which strengthens the performance of molecule prediction.

## K ANALYSIS OF COMPUTATIONAL COSTS

We observe that introducing feedback iterations indeed increases computational costs. During training, based on the same batch size, the memory is increased by around 0.9GB. However, in Fig. 8, we observe that the convergence speed is significantly faster than the baseline Zhuang et al. (2023).

During inference, compared with one-step disentanglement, the inference time of using 8 iterations is increased by around 0.06s. Although the inference time is longer, the disentangled invariant representations are more accurate, which is instrumental in improving the generalization ability. Particularly, for molecules with multiple atoms and various structures, we observe that employing

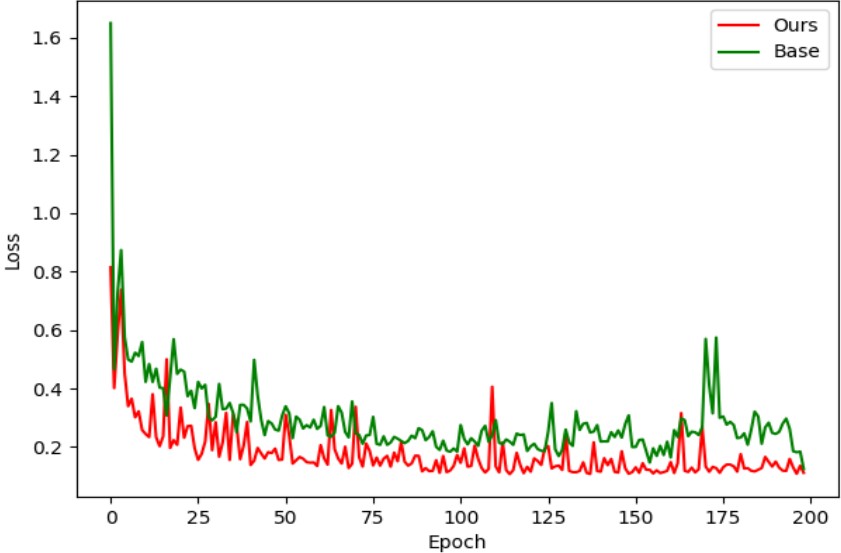

Figure 8: The training loss of the baseline Zhuang et al. (2023) and our method.

our method could significantly improve the performance of downstream tasks, further demonstrating the effectiveness of our method.

## L    QUANTIFY CONTRIBUTION OF FEEDBACK MECHANISM

To quantify the contribution of feedback iterations, we make an ablation experiment of the iteration number. Here, we only change the iteration number. Other operations are kept unchanged. The results are shown in Table 5.

Table 9: Quantify contribution of feedback mechanism on highly complex molecules.

| 1 | 4 | 8 | 12 | 16 |
|---|---|---|---|---|
| 0.3014 | 0.1867 | 0.1341 | 0.1279 | 0.1254 |

We can observe that compared with one-step disentanglement, using multiple feedback iterations indeed improves the performance, indicating that our method is beneficial for decomposing invariant molecule representations. Besides, we also observe that when the iteration number is larger than 12, the performance improvement is limited. The reason is that for the current dataset, using 12 iterations is sufficient for disentangling highly complex molecules.

Finally, we observe that complex molecules usually contain more atoms. Therefore, to further demonstrate the effectiveness of our method on highly complex molecules, we select those molecules with more than 12 atoms for additional evaluation. The results are shown in Table 9.

We can see that for complex molecules, using one-step disentanglement could not decompose invariant characteristics effectively. Using multiple feedback iterations could indeed decouple the representations involving plentiful invariant information, which strengthens the generalization ability.

## M    PROTEIN-LIGAND BINDING-AFFINITY PREDICTION

For OOD molecular representation learning, in Table 1 and 2, we show the corresponding results. Furthermore, in Table 3, our method is evaluated on an additional task, i.e., molecule's ground-state prediction. These results demonstrate that our method could effectively disentangle invariant molecule characteristics, which strengthens the generalization ability.

Finally, to further demonstrate the superiorities of our method, we evaluate our method on protein-ligand binding-affinity prediction. Binding affinity is an important metric in biology and medicinal chemistry that measures the strength of a reversible association between biological macromolecules,

such as proteins or DNA, and small-molecule ligands, such as drugs Boulougouri et al. (2024). Here, we follow the work Boulougouri et al. (2024) and employ the same datasets and metrics to verify our feedback disentanglement method. The results are shown in Table 10.

Table 10: Analysis of protein-ligand binding-affinity prediction.

| Method | RMSE | MAE | R |
|---|---|---|---|
| MAT | 1.457 | 1.154 | 0.747 |
| DimeNet | 1.453 | 1.138 | 0.752 |
| CMPNN | 1.408 | 1.117 | 0.765 |
| SR-BIND Boulougouri et al. (2024) | 1.383 | 1.122 | 0.780 |
| CFD | **1.356** | **1.073** | **0.794** |

Here, MAT, DimeNet, CMPNN are three GNN-based methods. **RMSE**, **MAE**, and **R** are three predefined metrics. We can observe that for the challenging protein-ligand binding-affinity prediction, employing our method still improves the performance for the given three metrics. This further demonstrates that for biological macromolecules, using feedback disentanglement is meaningful, which improves the generalization of the learned molecule representations.

# N    MORE INTERPRETATIONS OF KAN

In general, molecules consist of multiple atoms and bonds. For certain biomacromolecules, their structures are entanglement of various substructures, which poses the challenge for learning molecule concepts that focus on critical substructures.

Inspired by the Kolmogorov-Arnold representation theorem Kolmogorov (1961), KAN allows learning custom activations of the network. In this way, it is possible to analyze the contribution of individual components of the input, then providing a more transparent view of the network's decision-making process.

Thus, using KAN could sufficiently convert the analyzed components to the corresponding concepts that involve plentiful molecule-aware substructure information, which is beneficial for strengthening the generalization of molecule representations.

