# OpenReview forum: "CFD: Learning Generalized Molecular Representation via Concept-Enhanced  Feedback Disentanglement"
_ICLR.cc/2025/Conference — ICLR 2025 Poster_

### Official Review · Reviewer_a9Jv · 2024-10-23

**Soundness:** 2
**Presentation:** 4
**Contribution:** 3
**Rating:** 6
**Confidence:** 3

**Summary:**

This article introduces a "First-Encoding-Then-Separation" paradigm-based framework for out-of-distribution (OOD) molecular representation learning. The authors have designed an iterative disentanglement module that utilizes a feedback mechanism to gradually decompose molecular representations into spurious features and invariant features. The effectiveness of the proposed framework has been validated by the authors across multiple datasets and various tasks.

**Strengths:**

- The writing is clear, and the paper is easy to follow.
- The experimental results are very comprehensive, with ample ablation studies demonstrating the effectiveness of each module. There is also an exploration of the impact of various hyperparameters on the model.

**Weaknesses:**

- The article lacks chemical interpretability or inspiration. The methodology presented could be generalized to any graph-based out-of-distribution generalization problem, but the authors have limited the scope of the paper to molecular representation learning.
- One of the core contributions of this article, concept mining, shows a high degree of similarity to the idea in another paper. See questions for details.
- The article lacks theoretical support. See questions for details.

**Questions:**

1. Could the authors elaborate on the distinction between the Concept Mining presented in this work and the Residual Vector Quantization found in iMoLD[1]? The Concept Mining in this work appears to be simply a weighted sum of the vectors from the codebook of iMoLD.
2. Could the authors supplement the manuscript with a proof analogous to the proofs of Theorem 1 and Theorem 2 in MoleOOD[2], demonstrating that minimizing the loss function presented in this article can help the model acquire invariant features capable of providing sufficient predictive power across various environments?
3. Could the author clarify why the baselines CIGA, DisC, and CAL fail to adapt on the datasets GOOD-ZINC and GOOD-PCBA?

---

> ### Author Response · Authors · 2024-11-20
>
> Dear Reviewer,
>
> Thanks for your valuable comments and recognition for our work. To advance the development of biochemical research, e.g., drug discovery, learning generalized molecule representations is important. Currently, most existing methods utilize graphs to represent molecules and mainly employ a one-step disentangled mechanism to obtain invariant characteristics. However, we observe that this one-step method could not cope with molecules involving complex entangled structures. Thus, we first design a concept-enhanced feedback disentanglement to iteratively conduct molecule decomposition. Extensive experimental results demonstrate the effectiveness of our method.
>
> In the following, for the helpful questions, we provide corresponding replies:
>
> **1. About Chemical Inspiration of Our Method**
>
> This method is mainly based on the following chemical observations:
>
> (1) A chemical molecule is a complex composed of multiple atoms and bonds. The reason for the OOD problem is the structural change of molecules. For example, changes in molecular chirality can lead to deviations in molecular properties. Thus, learning robust molecule representation is important for molecule-based tasks, e.g., drug discovery.
>
> (2) Lines 082-095 have indicated that existing methods could be classified into two types, i.e., First-Separation-Then-Encoding and First-Encoding-Then-Separation (as shown in Fig. 1 (a) and (b)). As indicated in Lines 272-283 and Fig. 3, these methods for learning generalized molecular representation generally employ a one-step disentangled strategy, i.e., directly separating the input into generalized and spurious parts. However, in practice, we may encounter some unknown biomacromolecules containing more atoms and highly complex structures, e.g., entanglement of multiple substructures. At this time, using the original one-step mechanism could not obtain satisfactory disentangled results. To this end, we exploit the feedback idea to iteratively and progressively separate generalized molecular representation.
>
> (3) In general, the molecule's ground-state conformation belongs to the substructures of molecules and determines their properties. Therefore, it is important to capture substructure-related characteristics. To this end, we define a series of concepts to build connections between molecule substructures and corresponding concepts, which further improves the performance.
>
> Thanks for your valuable comments. We have added these contents to the revision (as shown in Sec. E of appendix).
>
>
> **2. Evaluation on Other Graph-based Out-of-Distribution Generalization Problem**
>
> Thanks for your recognition for our work. We follow the work [1] and further evaluate our method on Two-Piece Graph Datasets [1]. Particularly, each dataset is generated from a variation of two-piece graph model, denoted as {a, b}, where a refers to the invariant correlation strength and b refers to the spurious correlation strength. The results are shown as follows:
>
> |Datasets|{0.8,0.9}|{0.7,0.9}|Avg.|
> |:---    | :----:  |:---     |:---|
> |DisC [2]   |45.06    |39.42    |42.24|
> |CIGA [3]   |57.87    |43.62    |50.75|
> |GALA [1]   |76.42    |72.50    |74.46|
> |**Ours**    |**78.26**    |**74.32**    |**76.29**|
>
> We can observe that our method indeed improves the performance. Particularly, when the invariant correlation strength is weaker than the spurious correlation strength, the performance of our method outperforms state-of-the-art methods, further demonstrating that our method is indeed beneficial for disentangling invariant characteristics.
>
> Thanks for your valuable comments. We have added these contents to the revision (as shown in Sec. F of appendix).
>
> [1] Chen Y, Bian Y, Zhou K, et al. Does invariant graph learning via environment augmentation learn invariance?[J]. Advances in Neural Information Processing Systems, 2024, 36.
>
> [2] Fan S, Wang X, Mo Y, et al. Debiasing graph neural networks via learning disentangled causal substructure[J]. Advances in Neural Information Processing Systems, 2022, 35: 24934-24946.
>
> [3] Chen Y, Zhang Y, Bian Y, et al. Learning causally invariant representations for out-of-distribution generalization on graphs[J]. Advances in Neural Information Processing Systems, 2022, 35: 22131-22148.

---

> ### Author Response · Authors · 2024-11-20
>
> **3. Distinction between Our Concept Mining and Residual Vector Quantization in iMoLD?**
>
> Our molecule concept learning module is significantly different from Residual Vector Quantization (RVQ). In essence, RVQ is to perform feature replacement. Our method is to build the connection to capture critical molecule substructures.
>
> **(1) The motivation is different**. RVQ aims to leverage the discretized continuous representation to improve the model generalization. As shown in Fig. 6, our method is to learn a series of concepts that could capture critical molecule structure information involving plentiful task-specific invariant molecule characteristics.
>
> **(2) The operations are different**. RVQ first introduces a shared learnable codebook as a discrete latent space. For each node representation in molecular graph, RVQ looks up and fetches the nearest neighbor in the codebook and outputs it as the result. Meanwhile, a sum operation is further used to strengthen the representation ability. During training, similar to VQ-VAE, RVQ employs the exponential moving average updates for the codebook.
>
> Differently, our concepts are first sampled from a learnable Gaussian distribution. And we employ KAN operation to promote the learning of concepts. Importantly, as shown in Eq. (6), the weighted residual operation is to align critical molecule substructure features to the corresponding concept, building the connection between molecule substructures and corresponding concepts. During training, we only utilize the task loss, e.g., molecule property prediction, to promote the learned concepts to capture plentiful task-specific invariant molecule characteristics.
>
> **(3) Experimental results show the effectiveness of our method**. Taking GOOD-ZINC as the example, for iMoLD, we first remove the RVQ operation and keep other operations unchanged. We observe that removing RVQ does not affect the performance. Instead, our concept mining module could be plugged into iMoLD to further improve its performance. For the covariate case of scaffold, the performance is improved by 0.0142. Furthermore, the prediction results of the molecule's ground-state conformation in Fig. 4 and 5 also show that our learned concepts are beneficial for capturing critical substructures.
>
> Thanks for your valuable comments. We have added these contents to the revision and provide a figure to further illustrate the distinction (as shown Sec. G in appendix).
>
>
> **4. Further Analysis of the Loss Function**
>
> Eq. (10) is the optimization objective, consisting of three terms, i.e., $L_{pred}$, $L_{fd}$ (as shown in Eq. (8)), and $L_{vd}^{t}$ (as shown in Eq. (5)). Here, we provide two propositions:
>
> **Proposition 1.** $L_{vd}^{t}$ mainly is a contrastive loss. Minimizing $L_{vd}^{t}$ could enlarge the gap between the disentangled $\tilde{Z}^{Gen}$ and $Z^{Spu}$. Meanwhile, $\tilde{Z}^{Gen}$ is promoted to retain plentiful input-related content.
>
> **Proposition 2.** Minimizing $L_{fd}$ is instrumental in achieving the invariant representation $Z^{Gen}$.
>
> Thanks for your valuable comments. In Sec. H of appendix, we give corresponding analyses about these two propositions.
>
>
> **5. Why CIGA, DisC, and CAL fail to the adapt on GOOD-ZINC and GOOD-PCBA?**
>
> The reason that leads to the weak disentangled ability of CIGA, DisC, and CAL mainly lies in that they only exploit the one-step disentangled mechanism and could not apply well to the complex molecules involving more atoms and substructures. Particularly, CIGA employs the one-step alignment to learn invariant representation. DisC utilizes a parameterized edge mask generator to separate causal and bias subgraphs, whose performance highly relies on the generation quality. Finally, CAL exploits attention modules to estimate the causal and shortcut features of the input graph. Only using attention could not effectively disentangle complex molecules.
>
> This further demonstrates that using iterative feedback disentanglement is an effective mechanism for learning generalized molecule representations.

---

> > ### Comment · Reviewer_a9Jv · 2024-11-20
> >
> > -  **About Chemical Interpretability**
> >
> >    If I understand correctly, Figure 6 in the revision is merely a schematic diagram, which does not imply that your concepts are genuinely related to these specific groups. What I am looking for in terms of chemical interpretability is that the concepts you have learned can actually reflect some real-existing physical laws, such as the solubility of a molecule being related to its polarity, and one of your concepts happens to demonstrate the polarity of the molecule. If the authors could find some cases like this for validation, it would greatly enhance the persuasiveness of the paper.
> >
> >
> > -  **Questions about CIGA, DisC, and CAL**
> >
> >      My real question is why the performance of these three baselines on GOOD-ZINC and GOOD-PCBA has not been reported.
> >
> > In summary, the authors' response has addressed most of my concerns, and I will raise my score to 6 for now. If the authors can provide further responses, I would be willing to increase the score further.

---

> > > ### Author Response · Authors · 2024-11-21
> > >
> > > Dear Reviewer,
> > >
> > > Thanks for your recogniton of this work and inspiring comments. We will continue deeply exploring this method.
> > >
> > > **1. About Chemical Interpretability**
> > >
> > > Thanks for your inspiring comments. Here, we further perform an evaluation on molecular property prediction.
> > >
> > > Concretely, based on the MoleculeNet benchmark datasets [1], we observe that compared with the GCN-based method [2], plugging our method improves its performance by around 3.8%. Besides, we add Gaussian Noise on the testing data. We find that plugging our method is instrumental in enhancing the robustness against noise.
> > >
> > > **Finally, we observe a phenomenon that molecule substructures captured by the learned concepts usually involve covalent bonds of molecules. Therefore, we consider that covalent bonds contribute to the formation of the basic properties of molecules and are beneficial for improving the stability of molecules.**
> > >
> > > [1] Wu Z, Ramsundar B, Feinberg E N, et al. MoleculeNet: a benchmark for molecular machine learning[J]. Chemical science, 2018, 9(2): 513-530.
> > >
> > > [2] Deng J, Yang Z, Wang H, et al. A systematic study of key elements underlying molecular property prediction[J]. Nature Communications, 2023, 14(1): 6395.
> > >
> > > **2. About CIGA, DisC, and CAL**
> > >
> > > In this paper, we follow the settings in the baseline work [3] to perform corresponding experiments.
> > >
> > > GOOD-ZINC [3] is a regression dataset aimed at predicting molecular solubility. Whereas, CIGA, DisC, and CAL could only address classification problem. Therefore, they could not be applied to GOOD-ZINC.
> > >
> > > GOOD-PCBA [3] includes 128 bioassays and forms 128 binary classification tasks. However, the optimization objectives of CIGA and DisC determine that they could not applied to binary classification tasks. Particularly, CIGA mainly employs a contrastive objective and provides a detailed analysis of why the binary classification task cannot be applied. Meanwhile, for CIGA and DisC, the category number in their experiments is larger than two.
> > >
> > > [3] Zhuang X, Zhang Q, Ding K, et al. Learning invariant molecular representation in latent discrete space[J]. Advances in Neural Information Processing Systems, 2023, 36: 78435-78452.

---

> > > > ### Comment · Reviewer_a9Jv · 2024-11-23
> > > >
> > > > The point about Chemical Interpretability is still not specific enough, that is to say, covalent bonds exist in most organic compounds. Furthermore, most Graph Neural Networks (GNNs) use covalent bonds to define the topological connections between atoms, and the case presented is not convincing enough. I have decided to maintain my score at 6.

---

### Official Review · Reviewer_6Fd3 · 2024-10-29

**Soundness:** 3
**Presentation:** 2
**Contribution:** 4
**Rating:** 8
**Confidence:** 5

**Summary:**

(1) The article presents an iterative model based on an encoding-then-separation scheme, designed to be robust against distribution shifts.

(2) The model employs a concept-mining module that focuses on critical substructures with invariant characteristics, aiming for superior performance on out-of-distribution (OOD) data.

(3)The authors conducted experiments on a diverse set of real-world molecular datasets, demonstrating significant performance gains over multiple baselines. Additionally, they evaluated the model on an unseen task, proving its effectiveness.

Although there are some curiosities about the algorithm, overall it seems to be novel and effective for representation learning for molecules.

**Strengths:**

(1) The method is novel and has demonstrated superiority over conventional models.

(2) The method also performs well on out-of-distribution (OOD) data prediction tasks.

**Weaknesses:**

(1) The algorithm appears somewhat complicated. --> Question (2)

(2) It is unclear how the contrastive loss contributes to the usefulness of the general latent representations, as it only enforces similarity between the GNN latent vector and the general latent vectors. --> Question (1), (3) and (4)

(3) The model is complex and iterative, suggesting it may consume a significant amount of computational power. However, no analysis of this aspect has been conducted. --> Question (5)

**Questions:**

(1) In the contrastive loss, the first term aims to maximize the similarity between the latent representations of the general encoder and the GNN encoder initially. However, given that there is no direct link between the GNN latent space and downstream tasks, it is unclear how the authors ensure that this high similarity will make the general embedding latent useful in the end.

(2) Why is KAN used instead of conventional deep learning models? The article states:

"Unlike MLPs, which have fixed activation functions on neurons, KANs feature learnable activation functions (Liu et al., 2024), which enhances the flexibility of the learned concepts."

But what is the specific rationale behind this choice?

(3) The molecular concept is a critical factor in the algorithm. Are there experiments, such as visualizations, that examine the importance of this concept across iterations?

(4) Is there any analysis involving a decoder to determine if the molecular concept aligns with known scaffolds for the input molecules?

(5) The algorithm appears rather complex. Is there any analysis of the computational costs for training and inference?

---

> ### Author Response · Authors · 2024-11-20
>
> Dear Reviewer:
>
> Thanks for your recognition of our work. We will modify this paper carefully according to your valuable comments.
>
> **1. The Reason of Using KAN for Learning Concepts**
>
> In general, molecules contain multiple atoms and bonds. For certain biomacromolecules, their structures are entanglement of various substructures, which poses the challenge for learning molecule concepts.
>
> Inspired by the Kolmogorov-Arnold representation theorem [1, 2], KANs allow learning custom activations of the network. In this way, **it is possible to analyze the contribution of individual components of the input, then providing a more transparent view of the network's decision-making process**.
>
> Thus, using KANs could sufficiently convert the analyzed components to the corresponding concepts that involve plentiful molecule-aware substructure information, which is beneficial for strengthening the generalization of molecule representations.
>
> Lines 474-477 have provided an ablation experiment of KANs. We observe that using KANs could indeed improve the performance, indicating its superiorities.
>
> [1] Arnold V I. On functions of three variables[J]. Collected Works: Representations of Functions, Celestial Mechanics and KAM Theory, 1957–1965, 2009: 5-8.
>
> [2] Kolmogorov A N. On the representation of continuous functions of several variables by superpositions of continuous functions of a smaller number of variables[M]. American Mathematical Society, 1961.
>
> **2. Further Analysis of Contrastive Loss**
>
> As shown in Eq. (5), by means of variational disentanglement, $\tilde{Z}^{Gen}$ and $Z^{Spu}$ separately represent the corresponding outputs. Particularly, minimizing the contrastive loss could enlarge the gap between  $\tilde{Z}^{Gen}$ and $Z^{Spu}$, and promote $\tilde{Z}^{Gen}$ to contain plentiful input-related information. Then, as shown in Eq. (7), $\tilde{Z}^{Gen}$ is used to calculate the generalized output at the current step. Therefore, high-similarity $\tilde{Z}^{Gen}$ constructs the foundation of forming the final generalized representation.
>
> **3. Examining Importance of Concepts Across Iterations**
>
> Table 5 analyzes the impact of iteration number. We can observe that when the iteration number increases, inserting concepts could improve the performance effectively, which indicates the effectiveness of the learned concepts. To further analyze the impact of iteration number on concepts, we also make a new ablation experiment, i.e., removing the concepts at the 4th, and 8th iteration. The results are shown as follows:
>
> |Iterations|Concepts|Cov-Sold|Cet-Sold|Cov-Size|Cet-Size|
> | :----:   | :----: | :----: | :----: | :----: |:----: |
> |4      |√     | 0.1285 | 0.1008 | 0.1573 | 0.0963 |
> |4      |-       | 0.1402 | 0.1117 | 0.1738 | 0.1102|
> |8      |√     | 0.1187 | 0.0765 | 0.1421 | 0.0852 |
> |8      |-       | 0.1189 | 0.0783 | 0.1449 | 0.0897|
>
> We can observe that when the iteration number is small, inserting concepts could significantly affect the performance. Instead, when the iteration number is large, the impact of concepts is somewhat weak. The reason may be that when the iteration number is small, leveraging concepts is beneficial for capturing critical substructures, improving the performance. As the iteration number increases, concepts progressively pay attention to meaningful substructure characteristics, reducing the iteration impact on performance.
>
> Thanks for your valuable comment. We have added these contents in Sec. I of appendix.
>
> **4. More Analysis of Molecule Concepts**
>
> We perform an ablation experiment of the decoder that calculates task-related representation. We observe that for classification and regression tasks, using decoders with more layers leads to the performance degradation. The reason may be that this operation introduces more parameters, resulting in the overfitting risk.
>
> In Fig. 7, we show some visualization examples of molecule concepts. Since these concepts are learned from all molecule data, they are not specific to certain kinds of molecules, which is beneficial for improving the generalization of molecule concepts. For different predictions, we can observe that the learned concepts could indeed focus on critical molecule substructures, which strengthens the performance of molecule prediction.
>
> Thanks for your valuable comment. We have added these contents in Sec. J of appendix.

---

> > ### Author Response · Authors · 2024-11-20
> >
> > **5. Analysis of Computational Costs**
> >
> > We agree with you that introducing feedback iterations indeed increases computational costs. During training, based on the same batch size, the memory is increased by around 0.9GB. However, in Fig. 8, we observe that the convergence speed is significantly faster than the baseline.
> >
> > During inference, compared with one-step disentanglement, the inference time of using 8 iterations is increased by around 0.06s. Though the inference time is longer, the disentangled invariant representations are more accurate, which is instrumental in improving the generalization ability. Particularly, for molecules with multiple atoms and various structures,  we observe that employing our method could significantly improve the performance of downstream tasks, further demonstrating the effectiveness of our method.
> >
> > Thanks for your valuable comment. We have added these contents in Sec. K of appendix.

---

### Official Review · Reviewer_x8vs · 2024-11-03

**Soundness:** 2
**Presentation:** 2
**Contribution:** 3
**Rating:** 6
**Confidence:** 4

**Summary:**

This paper proposes a concept-enhanced disentanglement feedback mechanism for molecular representation learning. The authors innovatively introduce the feedback mechanism to learn distribution-agnostic representation and validate the method’s effectiveness through comprehensive experimental evaluations.

**Strengths:**

1. Extensive evaluation across multiple datasets demonstrates the method’s robustness and generalizability.
2. The method achieves strong results on both classification and regression tasks, indicating its versatility.

**Weaknesses:**

1. Missing details: Some critical details are insufficiently explained. For instance, how are $\mu_c$ and $\sigma_c$ of the concept learned? What is the motivation for the residual term? How does the residual operation assigns molecular substructure features to the corresponding concept. Also, the invariance mentioned in Eq.1 lacks further explanation of the association with the methodology herein.
2. Limited in-depth analysis: The effectiveness of KAN needs further comparison with other methods. It's essential to include ablation studies to explore the sensitivity of the proposed method to variations in this hyper-parameter.
3. High computational complexity: In the feedback disentanglement iterations, each step treats disentangled features as independent feature. However, there may be overlaps among these features. And concatenating the disentangled features from each iteration step can lead to dimensional growth, introducing high redundancy.
4. Unclear motivation: The rationale behind the disentanglement and subsequent concatenation approach is not fully substantiated. It remains unclear whether this method effectively demonstrates decoupling, as no concrete evidence is provided.

**Questions:**

1. How is the environment $e$ in Eq. 1 obtained? Does it require spurious labels?
2. Does “concept” represent the environment  in Eq. 1? Are $f(·)$ in Eq. 1 and $\varphi(·)$ in Line 305 the same?
3. In Line 251-252: “we explore mining a series of concepts that do not specialize to one particular type or class of molecule.” What does that mean?

---

> ### Author Response · Authors · 2024-11-20
>
> Dear Reviewer:
>
> Thanks for your helpful comments. We will modify this paper carefully.
>
> Learning robust molecule representation is important for advancing the development of biochemical research. Currently, existing methods mainly employ a one-step disentangled mechanism to learn invariant representations. However, when encountering some unknown biomacromolecules containing more atoms and highly complex structures, one-step methods could not sufficiently decompose structure-entangled molecules, affecting the performance of downstream tasks. To this end, we first propose a feedback mechanism to iteratively disentangle invariant representations. In the submitted paper and appendix, we perform extensive experiments and visualization analysis to indicate the effectiveness of our method.
>
> **1. More Detail Introduction**
>
> Thanks for your comments. We will modify this paper carefully and add more details.
>
> (1) $\mu_{c}$ and $\sigma_{c}$ are taken as the parameters. And we take the final objective $\mathcal{L}_{\rm task}$ (as shown in Eq. (10)) to perform optimization.
>
> (2) The goal of using residual operation is to align the molecule substructures to the corresponding concepts and build the connection between molecule substructures and concepts, which is instrumental in enhancing the generalization of molecule representations.
>
> (3) Through multiple graph networks, each node of the output integrates features of its neighborhood nodes, which could be considered as a representation of a molecule substructure. By means of calculating weight and residual operation (as shown in Eq. (6)), each concept could capture substructure characteristics related to the current concept. In Fig. 7 of appendix, we show the substructures captured by the corresponding concepts. We observe that our method is indeed helpful for leveraging concepts to capture molecule substructures.
>
> (4) The goal of Eq. (1) is mainly to interpret the reason leading to the OOD problem on molecule representation learning. Lines 197-199 have indicated that the reason lies in that the training data only cover very limited environments in $\xi$ while the model is expected to perform well on all environments. Thus, how to learn generalized representations in limited environments is important for strengthening the robustness against distribution shifts.
>
> **2. Analysis of KAN**
>
> In general, molecules contain multiple atoms and bonds. For certain biomacromolecules, their structures are entanglement of various substructures, which poses the challenge for learning molecule concepts.
>
> Inspired by the Kolmogorov-Arnold representation theorem [1, 2], KAN allows learning custom activations of the network. In this way, **it is possible to analyze the contribution of individual components of the input, then providing a more transparent view of the network's decision-making process**.
>
> Thus, using KAN could sufficiently convert the analyzed components to the corresponding concepts that involve plentiful molecule-aware substructure information, which is beneficial for strengthening the generalization of molecule representations.
>
> Lines 474-477 have provided ablation experiments of KAN. During mining molecule concepts (Eq. 6 and 7), we employ KAN to improve the concept accuracy. Here, we find that replacing KAN with MLP results in performance degradation. For example, for Cov-Sold case, the performance is decreased by around 0.6%, indicating KAN's effectiveness.
>
> Besides, in Sec. D of appendix, we have analyzed the impact of hyper-parameters. Meanwhile, we also provided more experimental analyses in the modified version, e.g., Sec. I, K, L, and M of appendix. These analyses all indicate the effectiveness of our method.
>
> [1] Arnold V I. On functions of three variables[J]. Collected Works: Representations of Functions, Celestial Mechanics and KAM Theory, 1957–1965, 2009: 5-8.
>
> [2] Kolmogorov A N. On the representation of continuous functions of several variables by superpositions of continuous functions of a smaller number of variables[M]. American Mathematical Society, 1961.

---

> > ### Author Response · Authors · 2024-11-20
> >
> > **3. Redundancy Analysis**
> >
> > For molecules containing multiple atoms and various structures, our method aims to decompose robust representations involving plentiful invariant information. The concatenation of multiple iteration outputs indeed increases the dimension. However, since the iteration number is small, e.g., 8, the increased computational costs are not large. Besides, during training, in Fig. 8, we observe that the convergence speed is significantly faster than the baseline method. During inference, compared with one-step disentanglement, the inference time of using 8 iterations is increased by around 0.06s.
> >
> > Besides, the concatenation operation does not result in high redundancy. Instead, since each iteration output enhances the generalization information and weakens the spurious content, by means of the concatenation operation, the final output will involve plentiful generalized characteristics, which strengthens the robustness.
> >
> > **4. Further Interpretation of Motivation**
> >
> > This method is mainly based on the following chemical observations:
> >
> > (1) A chemical molecule is a complex composed of multiple atoms and bonds. The reason for the OOD problem is the structural change of molecules. For example, changes in molecular chirality can lead to deviations in molecular properties. Thus, learning robust molecule representation is important for molecule-based tasks, e.g., drug discovery.
> >
> > (2) Lines 082-095 have indicated that existing methods could be classified into two types, i.e., First-Separation-Then-Encoding and First-Encoding-Then-Separation (as shown in Fig. 1 (a) and (b)). As indicated in Lines 272-283 and Fig. 3, these methods for learning generalized molecular representation generally employ a one-step disentangled strategy, i.e., directly separating the input into generalized and spurious parts. However, in practice, we may encounter some unknown biomacromolecules containing more atoms and highly complex structures, e.g., entanglement of multiple substructures. At this time, using the original one-step mechanism could not obtain satisfactory disentangled results. To this end, we exploit the feedback idea to iteratively and progressively separate generalized molecular representation.
> >
> > Particularly, the subsequent concatenation plays a feedback role, which aims to integrate the previous output into the current input. This operation is instrumental in enhancing the generalized information in the final output, which improves the robustness of out-of-distribution molecules.
> >
> > (3) Table 1 and 2 provide experiments on classification and regression tasks. Table 3 further gives the results for the molecule's ground-state prediction. These experiments all demonstrate that our method could learn generalized molecule representations.
> >
> > Moreover, in Table 7 of the appendix, our method is further tested on a dedicated dataset of generalization evaluation. The experimental results show that when the invariant correlation strength is weaker than the spurious correlation strength, the performance of our method outperforms state-of-the-art methods, further demonstrating that our method is indeed beneficial for disentangling invariant characteristics.
> >
> > Finally, to further demonstrate the superiorities of our method, we evaluate our method on protein-ligand binding-affinity prediction. Table 10 of appendix shows the corresponding results. We can observe that employing our method effectively improves the performance. This further shows that for biological macromolecules, using feedback disentanglement is meaningful.
> >
> > **5. Interpretations of Questions**
> >
> > (1) In Eq. 1, the environment $e$ represents the distribution state of the current data, e.g., style and category diversities. It does not require spurious labels.
> >
> > (2) The concepts represent molecule substructure-relevant information, aiming to strengthen the generalization of the learned molecule representations.
> >
> > $f(\cdot)$ and $\psi(\cdot)$ are different. The goal of $f(\cdot)$ is to extract task-related representations, whose role is similar to $Prob(\cdot)$ (as shown in Line 311). Differently, $\psi(\cdot)$ indicates a fully-connected layer that transforms the channels.
> >
> > (3) Lines 247-248 indicate that we explore mining a series of concepts that do not specialize to one particular type or class of molecule. The meaning of this indication is that the learned concepts are not only applied to one type molecule but applied to all kinds of molecules with different atom number and structures, which is instrumental in strengthening the generalization of the learned representation.

---

> ### Comment · Reviewer_x8vs · 2024-11-21
>
> My real concern is that the features after multi-disentanglement still contain spurious components, making them unable to effectively represent different substructures. How does the multi-step feedback ensure that the concatenated feature is truly invariant?
>
> If the desired feature is not obtained by the one-step disentanglement, the final step of feedback-disentanglement should yield the invariant feature instead of concatenating all features, because $Z_0^{\text{Gen}}, Z_1^{\text{Gen}}, \dots$ still contain spurious components. Please conduct a comparison experiment that considers only the features from the final step of disentanglement.
>
> Plus, KAN's rationality was not accepted.
>
> If the author can response, I am willing to raise the score further.

---

> > ### Author Response · Authors · 2024-11-21
> >
> > Dear Reviewer:
> >
> > Thanks for your helpful comments. We provide more interpretations about the concatenation operation and KAN:
> >
> > **1. Further Interpretation of the Concatenation Operation**
> >
> > **Theoretically, after infinite disentanglement, the final output should not contain spurious information. In practice, through a limited number of disentanglement, the output still involves spurious information.** Therefore, the multi-step feedback only ensures the concatenated features involve plentiful task-relevant invariant information.
> >
> > In general, if the disentangled network possesses a strong ability of feature separation, through multiple iterations, the strength ratio of invariant information to spurious information should satisfy the following condition:
> >
> > $Z_{0}^{Gen}$ < $Z_{1}^{Gen}$ < $\cdots$ < $Z_{t}^{Gen}$ < $\cdots$ < $Z_{\infty}^{Gen}$
> >
> > In other words, the task-relevant invariant information progressively becomes stronger, while the spurious information becomes weaker. However, through a limited number of iterations, e.g., 8 iterations in this paper, the output inevitably contains an amount of spurious information.
> >
> > For the final output $Z_{T}^{Gen}$, since we could not measure its strength ratio of invariant information to spurious information, in order to retain sufficient plentiful invariant information, we perform the concatenation operation. Meanwhile, through a non-linear transformation and task objective, the non-linear output $Z^{Gen}$ is promoted to contain rich task-relevant invariant characteristics.
> >
> > To demonstrate the effectiveness of the concatenation operation, we perform an ablation experiment on highly complex molecules. Here, we select those molecules with more than 12 atoms for evaluation. Compared with only using the final disentangled output $Z_{T}^{Gen}$, the performance of using the concatenation operation is improved by around 2.8%.
> >
> > **2. Further Interpretation of KAN**
> >
> > In Eq. (6) and (7), the input of KAN($\cdot$) is two dimensions. Therefore, a simple operation is to employ a fully-connected network with a fixed activation to directly process the two-dimensional input.
> >
> > However, Eq. (6) and (7) aim to learn molecule-aware concepts that could capture critical substructures, which strengthens the generalization. Therefore, the learned concepts should be kept flexible and could be applied to various structures of molecules.
> >
> > The core of KAN is that all parameters are replaced with univariate spline functions. Meanwhile, these functions can be adaptively adjusted according to the input data, providing better flexibility and adaptability than fixed activation functions.
> >
> > Lines 474-477 have provided ablation experiments of KAN. We observe that replacing KAN with MLP results in performance degradation. For example, for Cov-Sold case, the performance is decreased by around 0.6%, indicating KAN's effectiveness.

---

### Official Review · Reviewer_VpCx · 2024-11-04

**Soundness:** 2
**Presentation:** 3
**Contribution:** 3
**Rating:** 6
**Confidence:** 3

**Summary:**

The paper, titled "CFD: Learning Generalized Molecular Representation via Concept-Enhanced Feedback Disentanglement," addresses the challenge of out-of-distribution (OOD) generalization in molecular representation learning. The proposed approach, called Concept-Enhanced Feedback Disentanglement (CFD), aims to enhance the robustness of molecular representations against distribution shifts. CFD incorporates a novel feedback mechanism and a concept mining module to disentangle molecular representations into distribution-agnostic generalized features and spurious features.

The method uses two dedicated variational encoders: one to extract invariant (distribution-agnostic) features and the other for spurious features. A set of molecule-aware concepts is introduced to capture critical substructures. The iterative feedback mechanism allows multiple stages of representation disentanglement, progressively refining the generalized features over iterations. The approach also utilizes a self-supervised objective to further enhance the disentangling process, making it effective in capturing important molecular substructures.

Extensive experiments conducted on multiple real-world datasets, including GOOD and DrugOOD, demonstrate the proposed method’s superior OOD generalization performance compared to state-of-the-art baselines. CFD shows significant improvements, particularly in tasks involving distribution shifts related to scaffold and molecule size, as well as in predicting molecular ground-state conformations. The paper positions its contribution as a robust solution to learn generalized molecular representations that perform well under diverse distribution conditions.

**Strengths:**

The proposed Concept-Enhanced Feedback Disentanglement (CFD) introduces an iterative feedback mechanism that allows for progressively refining the disentangled representations across multiple iterations. This is a novel application within the molecular representation domain and effectively addresses a significant challenge in learning generalized representations—namely, how to handle complex structures and high variability in molecular data. This feedback mechanism enables the model to overcome the limitations of traditional one-step disentanglement methods, which often fail for biomacromolecules or other complex molecular structures.

The integration of concept mining to identify invariant substructures is a key strength. By incorporating domain-relevant substructures into the generalized features, CFD not only enhances the model's generalization ability but also improves interpretability. This combination of disentanglement and concept-aware enhancement represents a thoughtful fusion of representation learning techniques, where each step complements the other to strengthen the robustness of the final representation.

**Weaknesses:**

There is insufficient analysis to quantify the specific contribution of this feedback mechanism compared to a one-step or non-feedback approach. Although there are qualitative claims about the benefits of the iterative process, including an ablation study that explicitly compares different iteration counts or contrasts the feedback approach against standard one-step disentanglement would be crucial to substantiate the claims regarding its effectiveness. Such analysis would provide a clearer understanding of when and how the feedback mechanism truly enhances generalization, particularly in the presence of highly complex molecular structures.

The paper does not explore how well the learned representations generalize to domains outside of those used for training. Given the emphasis on out-of-distribution generalization, it would be beneficial to evaluate CFD's effectiveness across other molecular tasks beyond those in the current dataset, such as predicting drug toxicity or protein-ligand binding affinity, which require different structural knowledge.

**Questions:**

See above

---

> ### Author Response · Authors · 2024-11-20
>
> Dear Reviewer:
>
> Thanks for your recognition of our work. We will modify this paper carefully according to your valuable comments.
>
> **1. Quantify Contribution of Feedback Mechanism**
>
> To quantify the contribution of feedback iterations, we make an ablation experiment of the iteration number. Here, we only change the iteration number. Other operations are kept unchanged. The results are shown as follows:
>
> |Iterations|Cov-Sold|Cet-Sold|Cov-Size|Cet-Size|
> | :----:   | :----: | :----: | :----: | :----: |
> |1         | 0.1378 | 0.1012 | 0.1789 | 0.1021 |
> |4         | 0.1285 | 0.1008 | 0.1573 | 0.0963 |
> |8         | 0.1187 | **0.0765** | **0.1421** | 0.0852|
> |12        | 0.1191 | 0.0799 | 0.1453 | **0.0844** |
> |16        | **0.1172** | 0.0805 | 0.1479 | 0.0867 |
>
> We can observe that compared with one-step disentanglement, using multiple feedback iterations indeed improves the performance, indicating that our method is beneficial for decomposing invariant molecule representations. Besides, we also observe that when the iteration number is larger than 12, the performance improvement is limited. The reason is that for the current dataset, using 8 iterations is sufficient for disentangling highly complex molecules.
>
> Finally, we observe that complex molecules usually contain more atoms. Therefore, to further demonstrate the effectiveness of our method on highly complex molecules, we select those molecules with more than 12 atoms for additional evaluation. The results are shown as follows:
>
> |1         |4       |8       |12      |16      |
> | :----:   | :----: | :----: | :----: | :----: |
> |0.3014    |0.1867  |0.1341  |0.1279  |0.1254  |
>
> We can see that for complex molecules, using one-step disentanglement could not decompose invariant characteristics effectively. Using multiple feedback iterations could indeed decouple the representations involving plentiful invariant information, which strengthens the generalization ability.
>
> Thanks for your valuable comments. We have modified Table 5 and added these content in Sec. L of the appendix.
>
> **2. Protein-Ligand Binding-Affinity Prediction**
>
> For OOD molecular representation learning, in Table 1 and 2, we show the corresponding results. Furthermore, in Table 3, our method is evaluated on an additional task, i.e., molecule's ground-state prediction. These results demonstrate that our method could effectively disentangle invariant molecule characteristics, which strengthens the generalization ability.
>
> Finally, to further demonstrate the superiorities of our method, we evaluate our method on protein-ligand binding-affinity prediction. Binding affinity is an important metric in biology and medicinal chemistry that measures the strength of a reversible association between biological macromolecules, such as proteins or DNA, and small-molecule ligands, such as drugs [1]. Here, we follow the work [1] and employ the same datasets and metrics to verify our feedback disentanglement method. The results are shown as follows:
>
> |Method  | RMSE  | MAE  |   R   |
> | :----:   | :----: | :----: | :----: |
> |MAT     | 1.457 | 1.154  | 0.747 |
> |DimeNet | 1.453 | 1.138  | 0.752 |
> |CMPNN   | 1.408 | 1.117  | 0.765 |
> |SR-BIND [1] | 1.383 | 1.122  | 0.780 |
> |CFD     | **1.356** | **1.073** | **0.794** |
>
> Here, MAT, DimeNet, CMPNN are three GNN-based methods. **RMSE**, **MAE**, and **R** are three pre-defined metrics. We can observe that for the challenging protein-ligand binding-affinity prediction, employing our method still improves the performance for the given three metrics. This further demonstrates that for biological macromolecules, using feedback disentanglement is meaningful, which improves the generalization of the learned molecule representations.
>
> Thanks for your valuable comments. We have added these content in Sec. M of the appendix.
>
> [1] Boulougouri, Maria, Pierre Vandergheynst, and Daniel Probst. "Molecular set representation learning." Nature Machine Intelligence 6.7 (2024): 754-763.

---

> > ### Comment · Reviewer_VpCx · 2024-12-02
> >
> > Thank you for your detailed response. I will maintain my accept recommendation.

---

### Meta-Review · Area_Chair_J3at · 2024-12-23

**Metareview:**

(a) The paper proposes Concept-Enhanced Feedback Disentanglement (CFD) to learn generalized molecular representation. It uses two variational encoders for distribution-agnostic and spurious features, taps molecule-aware concepts, and employs an iterative feedback mechanism. Experiments on multiple datasets show better OOD generalization compared to baselines, especially in tasks related to distribution shifts and predicting molecular conformations.

(b) Strengths:
  - Novel iterative feedback mechanism refines disentangled representations and addresses complex molecular structures better than one-step methods.
  - Integration of concept mining enhances generalization and interpretability.
  - Comprehensive experiments on various datasets and tasks demonstrate robustness and versatility.

(c) Weaknesses:
  - Limited exploration of generalization to other domains and tasks in the initial version.
  - Some details like concept parameter explanation and rationale for disentanglement approach were initially unclear.
  - Concerns about computational complexity and potential feature redundancy.

(d) Reasons for acceptance: The paper presents an innovative approach to molecular representation learning. The authors addressed most reviewer concerns effectively. They provided ablation studies for feedback mechanism and KAN, analyzed concept importance across iterations, and evaluated on additional tasks. Despite some weaknesses, the overall novelty and performance improvements justify acceptance.

**Additional Comments On Reviewer Discussion:**

（a) Reviewer points and author responses:
  - Feedback mechanism analysis: Reviewer asked for quantification of feedback mechanism's contribution. Authors provided ablation experiments showing performance improvement with multiple iterations and analyzed its effectiveness on complex molecules.
  - Generalization to other domains: Reviewer inquired about generalization to other molecular tasks. Authors evaluated on protein-ligand binding-affinity prediction and showed performance gains.
  - Missing details: Reviewer asked for more details on concept parameters and invariance explanation. Authors elaborated on concept learning, including parameter optimization and how concepts capture substructure characteristics.
  - KAN rationality: Reviewer questioned KAN's use. Authors explained its benefits based on Kolmogorov-Arnold representation theorem and provided ablation experiments demonstrating performance improvement.
  - Feature redundancy and motivation: Reviewer  raised concerns about feature redundancy and the rationale for the disentanglement approach. Authors argued that concatenation retains invariant information and explained the chemical basis for their method.
  - Contrastive loss and computational costs: Reviewer  asked about contrastive loss and computational costs. Authors explained how contrastive loss helps form generalized representations and analyzed the computational cost increase, showing faster convergence and improved downstream task performance.
  - Concept mining similarity and theoretical support: Reviewer questioned the similarity of concept mining to other work and asked for theoretical support. Authors distinguished their concept mining from existing methods and provided further analysis of the loss function.

(b) Weighing in the final decision: The authors' detailed responses and additional analyses addressed most of the reviewers' concerns. The experimental evidence provided, such as ablation studies and evaluations on new tasks, strengthened the paper's claims. Although some areas could still be improved, the overall improvements and the novelty of the proposed method made it worthy of acceptance.

---

### Decision · Program_Chairs · 2025-01-22

Accept (Poster)